# Stochastic daily rainfall generation on tropical islands with complex topography

Lionel Benoit[1,2,5], Lydie Sichoix[2], Alison D. Nugent[3], Matthew P. Lucas[4], Thomas W. Giambelluca[1]

[1]Water Resources Research Center, University of Hawaiʻi at Mānoa, 96822 Honolulu, Hawaiʻi, USA
[2]GePaSud Laboratory, University of French Polynesia, 98702 Faaʻa - Tahiti, French Polynesia
[3]Department of Atmospheric Sciences, School of Ocean and Earth Science and Technology, University of Hawaiʻi at Mānoa, 96822 Honolulu, Hawaiʻi, USA
[4]Department of Geography, University of Hawaiʻi at Mānoa, 96822 Honolulu, Hawaiʻi, USA
[5]Now at Biostatistics and Spatial Processes (BioSP), INRAE, 84914 Avignon Cedex 9, France

*Correspondence to*: Lionel Benoit (benoitlionel2@gmail.com)

**Abstract.** Stochastic rainfall generators are probabilistic models of rainfall space-time behavior. During parameterization and calibration, they allow the identification and quantification of the main modes of rainfall variability. Hence, stochastic rainfall models can be regarded as probabilistic conceptual models of rainfall dynamics.

As with most conceptual models in Earth Sciences, the performance of stochastic rainfall models strongly relies on their adequacy in representing the rain process at hand. On tropical islands with high elevation topography, orographic rain enhancement challenges most existing stochastic models because it creates localized precipitations with strong spatial gradients, which break down the stationarity of rain statistics.

To allow for stochastic rainfall modeling on tropical islands, despite non-stationarity of rain statistics, we propose a new stochastic daily multi-site rainfall generator specifically for areas with significant orographic effects. Our model relies on a preliminary classification of daily rain patterns into rain types based on rainfall space and intensity statistics, and sheds new light on rainfall variability at the island scale. Within each rain type, the distribution of rainfall through the island is modeled by combining a non-parametric resampling of past analogs of a latent field describing the spatial distribution of rainfall, and a parametric Gamma transform function describing rain intensity.

When applied to the stochastic simulation of rainfall on the islands of Oʻahu (Hawaiʻi, United States of America) and Tahiti (French Polynesia) in the tropical Pacific, the proposed model demonstrates good skills in jointly simulating site-specific and island-scale rain statistics. Hence, it provides a new tool for stochastic impact studies in tropical islands, in particular for watershed water resources management.

## 1 Introduction

Stochastic rainfall generators are probabilistic tools aiming at simulating synthetic rainfall that mimic as closely as possible the statistical signature of rain observations [*Richardson*, 1981] [*Wilks and Wilby*, 1999] [*Ailliot et al.*, 2015]. More specifically, stochastic rainfall modeling consists of statistical learning (i.e., inference) of the joint space-time probability

density function (pdf) of rainfall at all sites and times of interest, and sampling this pdf to efficiently generate synthetic rainfall.
The ability of stochastic rainfall generators to emulate long and realistic rainfall sequences makes them an appropriate tool for
the simulation of design storms [*Niemi et al.*, 2016]. Simulated rainfall can then be used as inputs for impact models assessing
the effects of rainfall on different environmental processes including hydrology [*Paschalis et al.*, 2014], water resources
[*Cappelaere et al.*, 2020], geomorphology [*Peleg et al.*, 2020], and agronomy [*Mavromatis and Hansen*, 2001]. The
probabilistic approach followed by stochastic rainfall generators enables a comprehensive study of rainfall variability and, in
turn, the assessment of uncertainty propagation along the whole modeling chain [*Gabellani et al.*, 2007]. This makes stochastic
rainfall generation a key tool for management of rain-induced risk, in particular, for flood [*Caseri et al.*, 2016] and drought
risks [*Supit et al.*, 2012]. In addition, the focus of stochastic rainfall models on the statistical signature of rainfall creates new
ways to characterize rainfall space-time behavior [*Marra and Morin*, 2018], and assess the impact of rainfall variability on the
hydrosphere [*Morin et al.*, 2019]. Finally, when conditioned to climate model outputs, stochastic rainfall generation can be
used for the downscaling of future precipitation projections [*Maraun et al.*, 2010], resulting in local-scale and high-resolution
scenarios of the possible evolution of rainfall in the context of climate change [*Jha et al.*, 2014] [*Volosciuk et al.*, 2017].
To enable fast and computationally efficient simulations, and thereby allow for investigation of rainfall variability
and associated uncertainty through the simulation of large ensembles, stochastic rainfall generators adopt an empirical
approach that bypasses the detailed physical modeling of rain generation processes [*Bauer et al.*, 2015]. To avoid the pitfall of
physically unrealistic simulations, stochastic rainfall models embed a significant part of our conceptual knowledge about
rainfall behavior in their parameterization (i.e., they implement statistical relationships that reflect as closely as possible the
physical processes at work). However, rainfall properties [*Krajewski et al.*, 2003] and, in turn, the performance of stochastic
rainfall generators [*Breinl et al.*, 2017] [*Vu et al.*, 2018] strongly depend on the climate of the area of interest. Hence, different
models have been proposed for different climates with each model focusing on a specific aspect of rainfall, for instance: rainfall
seasonality in monsoonal climates [*Greene et al.*, 2011]; rainfall spatial-temporal correlation in temperate climates [*Paschalis*
*et al.*, 2013]; or rainfall occurrence and extreme intensities in arid regions [*Wilcox et al.*, 2021].
On high tropical islands, or islands with high elevations and significant topography, rainfall is strongly location
dependent due to complex interactions between atmospheric circulation and island topography, which trigger different
mechanisms of orographic rain enhancement [*Foresti and Pozdnoukhov*, 2012] [*Houze*, 2012]. At monthly to annual scales,
the effect of orographic lifting of relatively steady trade winds generates well defined rain patterns [*Lyons*, 1982]. In these
patterns, highlands are usually wetter than lowlands, windward slopes wetter than leeward sides, and in case of successive
mountain ridges the first to be reached by the wet air masses is the wettest [*Giambelluca et al.*, 2013] [*Laurent et al.*, 2019].
To this first order quasi-static picture is added the important variability of daily rainfall patterns associated with processes
ranging from synoptic-scale disturbances [*Hopuare et al.*, 2018] [*Longman et al.*, 2021] to large-scale atmospheric circulations
[*Hopuare et al.*, 2015] [*Frazier et al.*, 2018] [*Brown et al.*, 2020]. Large deviations from the long term rainfall patterns are
thus observed, and usually dry leeward slopes can become the wettest part of the island, for example during Kona storms
(seasonal cyclones) in the Hawai'i archipelago [*Caruso and Businger*, 2006].

Orographic effects lead to non-stationary (i.e., non-homogeneous) rain statistics in both space and time, which challenges most existing stochastic rainfall models [*Nerini et al.*, 2017]. In the context of high tropical islands, the first difficulty arises from the long term patterns of orographic rain enhancement that create non-stationarities in space. Rainfall generators must therefore account not only for the correlation between locations [*Leblois and Creutin*, 2013] [*Paschalis et al.*, 2013], but also for the location of the rain within the island. When orographic effects are directly related to topography, Generalized Linear Models (GLMs) have been leveraged to account for spatial rain patterns by linking model parameters to local topographic information such as altitude or slope aspect [*Ambrosino et al.*, 2014] [*Chandler*, 2020]. In tropical islands however, the complexity of the rain-topography relationships (e.g., successive mountain ridges becoming drier at similar altitude) hinder the direct regression of model parameters on topographic data. An interesting option to overcome the non-uniqueness of topography-rainfall relationships is to interpolate the parameters of the rainfall model in space in order to let data inform non-stationarity [*Kleiber et al.*, 2012] [*Bennett et al.*, 2018]. This however increases model complexity, and therefore requires training datasets with high spatial resolution, which are not yet available in most target areas [*Benoit et al.*, 2021]. Finally, the difficulty of accounting for orographic effects in the context of high tropical islands is further increased by the temporal variability of rainfall patterns (e.g., wet leeward slopes during Kona storms), which calls for the development of models able to identify and capture the main modes of variability of orographic effects over time.

A rainfall generator compatible with marine tropical climates and complex rain-topography interactions encountered in tropical islands is, to the best of our knowledge, still missing in the toolbox of stochastic rainfall generators. To fill this gap, the present paper proposes a new rainfall model dedicated to high tropical islands with significant and complex topography, which aims to account for both the long-term quasi-static patterns of rain accumulation and the day-to-day fluctuations of the rainfall spatial distribution. Specifically, the goal is to develop a daily resolution multi-site stochastic rainfall generator able to simulate: (1) site specific rain occurrence, persistence, intensity and seasonality; (2) spatial patterns of daily rain accumulation; and (3) areal rain statistics at the island scale.

To achieve these objectives, the remainder of the article is structured as follows. Section 2 briefly reviews the main features of tropical island rainfall and describes our stochastic rainfall model. Section 3 illustrates the performance of the model for the island of Oʻahu (Hawaiʻi, USA) in the tropical Pacific, and a similar test study is repeated in supplementary material for the island of Tahiti (French Polynesia) to demonstrate the versatility of the model. Finally, section 4 discusses how the focus on orographic rain enhancement has influenced the design of the model and provides concluding remarks.

## 2 Data and methods

### 2.1 Rainfall features of interest

Because stochastic rainfall models are data-driven, their structure depends on the rain features one wants to reproduce in simulations. Hence, the identification of the main features of daily rainfall in high tropical islands is a prerequisite for the design of the present model. For illustration purposes, we focus throughout the main text on the island of Oʻahu, Hawaiʻi (lon

= 158°W, lat = 21.5°N, area = 1545 km$^2$, max altitude = 1220 m). The available rain gauge observation dataset consists of daily records from a network of 86 rain gauges spread over the island (Fig. 1a), and covers a 20-year period 1991–2011. It corresponds to a compilation of quality controlled and gap-filled daily observations [*Longman et al.*, 2018]. Gap-filling was performed using the normal ratio method [*Paulhus and Kohler*, 1952], and only stations with less than 5% gap-filled data are kept for this study in order to minimize the impact of gap-filling on our results. To contextualize the observed rain patterns, several meteorological covariates (e.g., pressure, temperature, humidity and wind) are investigated at the island scale. We use the ERA5 reanalysis dataset [*Hersbach et al.*, 2018] at 12:00 PM HST to inform these covariates and average the values of the 12 grid cells (pixel size = 0.25° x 0.25°) encompassing the island of Oʻahu.

Figure 1 displays the main features of rainfall over the island of Oʻahu. It shows the strong impact of trade wind induced orographic rain enhancement on the spatial distribution of annual precipitation (Fig. 1a), with windward (northeast) sides significantly wetter than leeward (southwest) ones, and highlands generally wetter than lowlands. Note some important details of this annual rainfall pattern, for example that the rain maximum is observed leeward of the main crest of the Koʻolau range, and that the Koʻolau mountains are significantly wetter than the Waiʻanae range despite higher elevation. In addition to prevailing orographic rainfall triggered by the interactions of trade winds with island topography (east-northeasterly trade winds blow more than 280 days per year over the Hawaiian archipelago [*Longman et al.*, 2015]), the island of Oʻahu also experiences spatially widespread rain events, mostly triggered by regional atmospheric disturbances such as cold fronts originating from mid-latitudes and Kona storms (seasonal cyclones) [*Longman et al.*, 2021]. These atmospheric disturbances mostly occur during (boreal) winter, which corresponds to the local rainy season (spanning from October–March). They represent the main source of precipitation for dry leeward locations and are responsible for the enhanced seasonality of rain accumulation in these areas (Fig. 1a).

The diversity of rain generation mechanisms (e.g., orographic lifting, cold fronts, or Kona lows) coupled with the steep island topography of volcanic origin result in a complex distribution of rainfall in space and time, which produces highly variable island-scale rain statistics (i.e., statistics summarizing rain behavior throughout the island for a given day). Figure 1 b–d shows that at the scale of the island of Oʻahu, daily rainfall is strongly intermittent in space (only 3% of the days record rain at all gauge locations, and half of the time at least 20% of the gauges measure no rain, Fig. 1b), highly skewed (island-scale rain accumulation average < 2.25mm/day 50% of the time, but island-scale maximum accumulation > 15mm/day 50% of the time and reaches 500 mm/day, Fig 1c), and strongly variable in space (coefficient of variation > 1.3 50% of the time, and > 2.9 10% of the time).

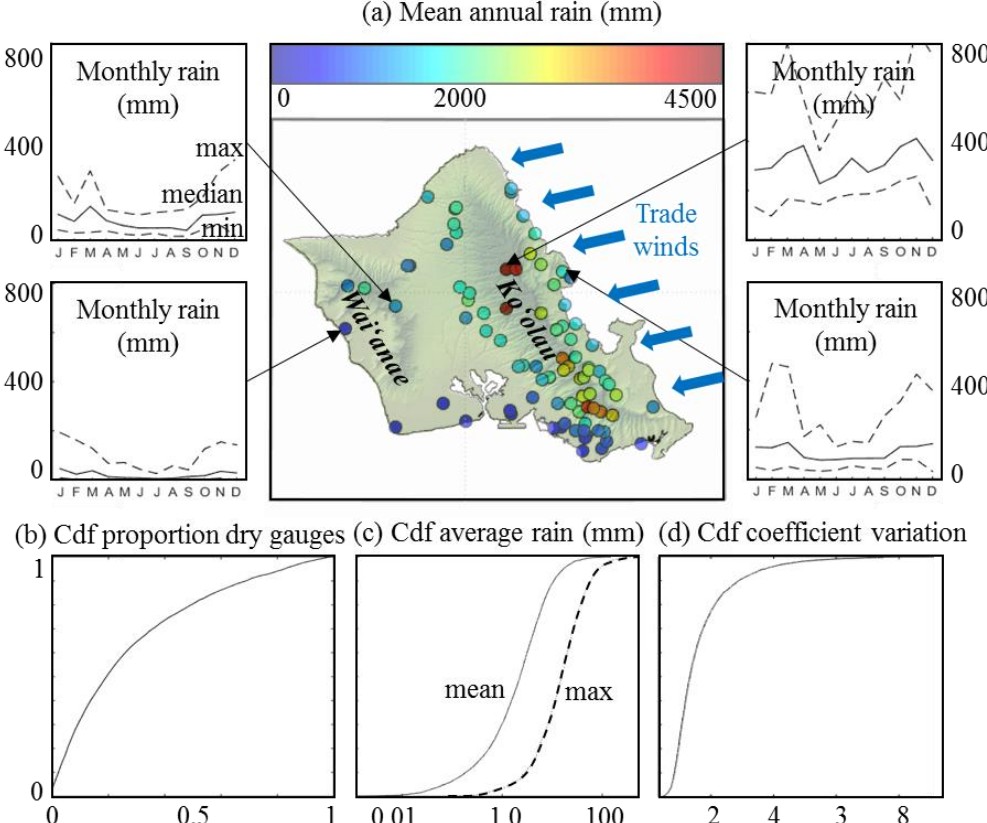

**Figure 1: Main features of rainfall observed over the island of Oʻahu.** (a) Mean annual rainfall (central panel) and seasonality of rain accumulation for four specific rain gauges (outer panels). (b) Cumulative distribution function (cdf) of the proportion of gauges measuring no rain for a given day. (c) Cdf of the mean and maximum daily rain accumulation computed over the whole observation network (abscissa is in log-scale). (d) Cdf of the coefficient of variation (i.e., standard deviation/mean) of daily rain accumulation throughout the rain gauge network.

## 2.2 Model description

### 2.2.1 Model overview

To model the statistical features of daily rainfall in tropical islands while accounting for the main mechanisms of orographic rain enhancement, in particular the variability of rainfall-topography relationships under the influence of changing atmospheric conditions, the proposed model splits rainfall behavior into three components: temporal variability, intensity (i.e., marginal distribution), and spatial distribution. Figure 2 summarizes the structure of the model, which is briefly introduced in the later part of this subsection, and will be discussed in detail in sections 2.2.2 to 2.2.4.

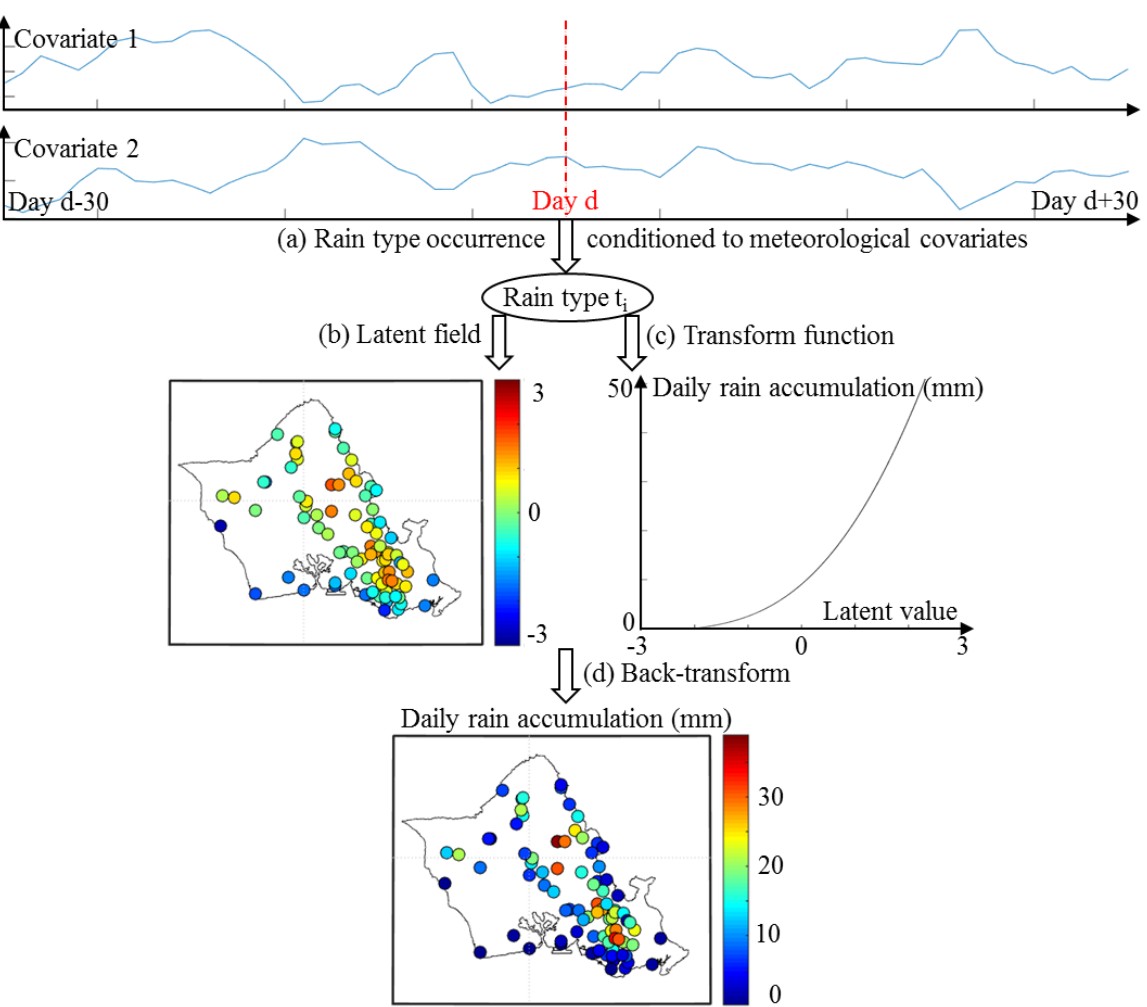


**Figure 2: Overview of the structure of the stochastic rainfall model.** (a) Meteorological covariates (here Geopotential at 950hPa and
Temperature difference between 950hPa and 700hPa) driving the occurrence of rain types, which summarize daily rain statistics. (b) Latent
field modeling of the spatial distribution of rainfall across the island. (c) Transform function linking latent values with actual rain
accumulations. (d) Back-transform combining (b) and (c) to obtain daily rain simulations.

The temporal variability of rain statistics and its relationships with the state of the atmosphere are modeled following

a rain typing approach (Fig. 2a) [*Ailliot et al.*, 2015] [*Benoit et al.*, 2018b]. In this framework, days with similar rain statistics
are first pooled together in a finite number of rain types, which represent summaries of island-scale daily rain statistics. Next,
rain type occurrence is modeled by a Markov chain describing how rain types transition to each other [*Bárdossy and Plate*,
1991] [*Wilby*, 1994]. To preserve climatological consistency and model the influence of atmospheric circulation on orographic
rain enhancement, rain type occurrence is conditioned to local meteorological covariates [*Benoit et al.*, 2020] making the
Markov chain non-homogeneous [*Hughes and Guttorp*, 1999] [*Vrac et al.*, 2007]. It is important to note here that we do not

use the widespread Hidden Markov Model (HMM) approach [*Ailliot et al.*, 2009] [*Greene et al.*, 2011] to introduce rain types in our stochastic rainfall model, but rather resort to a two-steps approach in which rain types are first determined by a direct classification of rain gauge observations, and subsequently a statistical rainfall model is built for each rain type. This choice has been motivated by the use of non-parametric methods to model the distribution of rainfall conditional to rain types, which prevents the formulation of the likelihood of the full statistical model and in turn the use of a HMM to model rain types.

Conditional to each rain type, the distribution of rain across the island is modeled following a meta-Gaussian approach (also referred to as trans-Gaussian or transformed Gaussian) [*Allard and Bourotte*, 2015] [*Baxevani and Lennartsson*, 2015] [*Papalexiou and Serinaldi*, 2020]. In this framework, a latent field with standardized Gaussian marginal distribution is non-linearly transformed to match the marginal distribution of daily rainfall across the island, and the spatial dependencies of the latent field are used to encode the spatial distribution of rainfall (Fig. 2b-c). The latent field is often assumed to follow a multivariate-Gaussian distribution, which allows for a parsimonious modeling of the spatial dependencies using geostatistics [*Lantuéjoul*, 2002], and in turn the spatial interpolation of rain gauge observations [*Benoit et al.*, 2018a]. In this study however, a careful investigation of inter-gauges spatial dependencies (Supplementary material 1) shows the inadequacy of the multivariate-Gaussian distribution to model the latent field. Instead, a non-parametric resampling of past analogs [*Gangopadhyay and Clark*, 2005] [*Yiou*, 2014] of the daily latent field is used to model spatial dependencies. Aside from the choice of the marginal distribution of the latent field (Gaussian vs. uniform), this model is comparable to the use of empirical spatial copulas [*Bárdossy and Pegram*, 2009] coupled with a parametric marginal distribution of non-zero rain accumulation. This approach has the advantage of faithfully reproducing the complex spatial distribution of rainfall due to orographic effects, but at the expense of the ability to spatial interpolation. The present model should therefore be regarded as a multi-site stochastic rainfall generator (and not a rain field generator), and the spatial interpolation of the simulated multi-site rainfall is kept out of the scope of this paper.

2.2.2 Meta-Gaussian representation of island-scale daily rainfall

Rain intensity and spatial distribution are modeled jointly following a meta-Gaussian approach (Fig. 2b-c). For a given day, the observed rain accumulations $R_{i=1 \dots N_T}$ across a network of $N_T$ gauges are linked to their latent counterparts $Z_i$ (which follow a standardized Gaussian marginal distribution, i.e., $Z \sim \mathcal{N}(0,1)$) through a non-linear transform function $\psi$. This transformation is performed by first assuming that non-zero rain accumulations observed throughout the island in a given day follow a Gamma distribution:

$$Z_i = \psi(R_i) = \Phi^{-1}\left(\frac{N_d}{N_T} + \frac{N_w}{N_T} \times Gamma(R_i; k, \theta)\right) \; if \; R_i > 0 \tag{1}$$

where $N_d$, $N_w$ are the number of dry and wet gauges, $\Phi^{-1}$ is the inverse cumulative distribution function (cdf) of the univariate standardized Gaussian distribution, and $Gamma(R_i; k, \theta)$ is the cdf of the Gamma distribution with shape parameter $k > 0$ and scale parameter $\theta > 0$.

In many instances, gauges measuring no rain (i.e., $R_i=0$) represent a significant part of the network, which creates a
concentration of zero values in rain accumulation distribution, and prevents a correct Gaussian transform using the function of
Eq. (1). To circumvent this problem, the latent values corresponding to dry gauges are assigned based on the distance of the
dry gauges to the closest wet gauge, such as the marginal distribution of the latent values matches the left portion of a
standardized normal distribution. The idea behind this application-specific solution to deal with the spatial intermittence of
rainfall is that a location far from any wet gauge should remain dry even after combining the associated latent field with slightly
different parameters of the transform function (Eq. 1-2) during the simulation step (Sect. 2.3.3). In case of a gauge measuring
no-rain ($R_i = 0$), the corresponding latent value is given by:
$$Z_i = \psi(R_i) = \Phi^{-1}\left(\left(1 - \frac{Dw_i}{max_{j=1:N_d}(Dw_j)}\right) \times \frac{N_d}{N_T}\right) \; if \; R_i = 0 \tag{2}$$

where $Dw_i$ is the distance of the gauge $i$ observing no rain to the closest gauge measuring non-zero rain. This transformation
has the advantage of creating spatial patterns of censored latent values (i.e., corresponding to dry gauges) that are coherent
with the ones of non-censored latent values (i.e., corresponding to wet gauges), and create smooth transitions between wet and
dry domains.
Once latent values ($Z_i$) are derived from rain observations ($R_i$), the spatial distribution of rain across the island is
defined by the spatial distribution of the latent field [*Bárdossy and Pegram*, 2009], i.e., the joint cdf of $Z_i$. As mentioned in
section 2.1, the spatial distribution of daily rainfall in high tropical islands is complex and strongly non-stationary due to
orographic effects, which prevents the use of a simple parametric form (such as the multivariate Gaussian distribution used in
most meta-Gaussian models of precipitation [*Benoit et al.*, 2018a] [*Papalexiou and Serinaldi*, 2020]) for the spatial
dependencies. Hence, in the present case, the non-parametric resampling of past latent fields is used to simulate the spatial
distribution of rainfall [*Rüschendorf*, 2009].

### 2.2.3 Rain typing

Based on the above meta-Gaussian representation of daily rain fields, days with similar rain statistics are pooled into
rain types (Fig. 2a) using a non-supervised clustering applied on the six-dimensional feature-space defined by the following:
- The three parameters of the transform function ($\psi$) (i.e., $p_0 = \frac{N_d}{N_T}$, k, θ), which inform the marginal distribution of daily
rainfall over the island.
- The first three components of the Karhunen-Loève expansion [*Huang et al.*, 2001] of the latent field Z
($PC_1, PC_2, PC_3$), which inform the spatial distribution of rainfall across the island.
Based on this feature-space $\boldsymbol{Y} = (p_0, k, \theta, PC_1, PC_2, PC_3)^T$, the clustering is performed using a Gaussian Mixture
Model (GMM, [*Fraley and Raftery*, 2002]) which approximates the pdf of $\mathbf{Y}$ as a weighted sum of multivariate Normal
distributions:
$$p_Y(\mathbf{Y} = \mathbf{y}) = \sum_{l=1:N_C} b_l \times \mathcal{N}(\mathbf{y} | \boldsymbol{\mu}_l, \boldsymbol{\Sigma}_l) \tag{3}$$

where $p_Y$ is the joint pdf of the random vector $Y$, $N_C$ is number of components in the GMM, $b_l$ is a weight assigned to the $l^{th}$ component, and $\mu_l$ and $\Sigma_l$ are the mean vector and covariance matrix of the multivariate normal distribution of the $l^{th}$ component. Here, the parameters embedded in the vector $Y$ are assumed to be only slightly correlated and the covariance matrices ($\Sigma_l$) are therefore assumed to be diagonal. The number of components of the GMM ($N_C$) is selected by minimization of the Bayesian Information Criterion (BIC [*Schwartz*, 1978]) estimated for different numbers of components in order to select a parsimonious classification (i.e., with as few rain types as possible) while properly fitting the pdf of $Y$ (i.e., $p_Y$). Once the pdf $p_Y$ is known, the probability that an observed vector $y_{obs}$ belongs to the $l^{th}$ component $C_l$ is given by:

$$p(y_{obs} \in C_l) = \frac{b_l \times \mathcal{N}(y_{obs} | \mu_l, \Sigma_l)}{\sum_{k=1}^{N_C} b_k \times \mathcal{N}(y_{obs} | \mu_k, \Sigma_k)} . \qquad (4)$$

And the classification is obtained by assigning each day ($d_i$) with a rain type (RT) that corresponds to the most probable mixture component:

$$RT(d_i) = max_{l \in 1..N_C}(p(y_i \in C_l)). \qquad (5)$$

### 2.2.4 Rain type occurrence

Once rain types have been defined based on rainfall statistical properties, their occurrence is conditioned to the vector $MC_d$ of meteorological covariates observed at day $d$ (Fig. 2a), and rain type occurrence is modeled by a non-homogeneous Markov Chain of order 1 [*Hughes and Guttorp*, 1999] [*Vrac et al.*, 2007]:

$$p(RT_d = j | RT_{d-1} = i, MC_d) \propto \gamma_{ij} \exp\left(-\frac{1}{2}(MC_d - \mu_{ij})\Sigma_{ij}^{-1}(MC_d - \mu_{ij})^T\right) \qquad (6)$$

Where $RT_d$ is the state of the Markov chain (i.e., the rain type) at day $d$, $p(RT_d = j | RT_{d-1} = i, MC_d)$ is the probability to transition from rain type $i$ to rain type $j$, $\Sigma_{ij}$ and $\mu_{ij}$ are the covariance matrix and the mean vector of the meteorological covariates when the transition from type $i$ to type j occurs, and $\gamma_{ij}$ is the baseline (i.e., long term average) probability of transition from type $i$ to type $j$. This model allows the transition probability $p(RT_d = j | RT_{d-1} = i, MC_d)$ to vary proportionally to the conditional density of $MC_d$ given the transition, and conditions the occurrence of rain types to the state of the atmosphere characterized by the covariates.

### 2.3 Model implementation

### 2.3.1 Selection of meteorological covariates

The set of meteorological covariates used for the conditioning of the non-homogeneous Markov Chain must be chosen so that: (i) the covariates are only weakly correlated to each other, which ensures model parsimony (i.e., minimal redundancy between covariates); and (ii) the temporal variations of the covariates are correlated with variations in rain type occurrence (Supplementary Material 2), which indirectly informs the seasonality and interannual variability of rainfall patterns. Note that in the present framework, the conditioning to covariates (i.e., the non-homogeneous part of the Markov chain) is used to inform the low frequency fluctuations of rain type occurrence (seasonal to interannual time scales), with higher frequencies (weekly

to daily time scales) being informed by the baseline transition probabilities ($\gamma_{ij}$). Hence, meteorological covariates are aggregated at the monthly scale prior to use for the conditioning of the non-homogeneous Markov chain. The monthly-aggregated covariates inform monthly anomalies in atmospheric conditions and, in turn, the likelihood of rain types to occur during a given month.

In the present case, we selected the meteorological covariates according to our initial knowledge about rain generation mechanisms in high tropical islands, and their links with the state of the atmosphere [*Elison Timm et al.*, 2014] [*Réchou et al.*, 2019] [*Sanfilippo*, 2020]; this led to the following five covariates.

1) Geopotential height at 700 hPa ($m^2.s^{-2}$). This covariate is correlated with the presence of synoptic-scale weather systems at the vicinity of the island and identifies regional atmospheric disturbances.

2) Temperature difference between 950 hPa and 700 hPa (K). This covariate is correlated with the lower atmospheric instability and identifies days prone to shallow convection.

3) Specific humidity at 700 hPa ($kg.kg^{-1}$). This covariate informs the presence of humidity above the height of the trade wind inversion and is negatively correlated with the strength of the inversion and positively correlated with the potential for deep convection and cold rain.

4) Meridional and 5) longitudinal humidity fluxes at 950 hPa (i.e., specific humidity multiplied by the *u* (east-west) or *v* (north-south) components of the wind field, $m.s^{-1}.kg.kg^{-1}$). These covariates provide the amount of moisture crossing over the mountain barrier available for precipitation and are a proxy for orographic precipitation.

**2.3.2 Model calibration**

The model is calibrated from a training dataset made of *N* days of rain accumulation recorded by a network of $N_T$ rain gauges (Fig. 1a). Data must be available for all stations and all days of the calibration period, and a preliminary gap-filling step is required in case of incomplete data [*Longman et al.*, 2018] [*Oriani et al.*, 2020]. Once a complete training dataset is available, the first step of model calibration consists of inferring the parameters of the transform function ($\psi$) for each day of the training period. This is performed by calculating the proportion of dry gauges and then estimating the parameters of the gamma distribution of the wet gauges using a maximum likelihood approach. Once the three parameters of $\psi$ are known, this function can be inverted to derive the latent values at each gauge location.

After calibration of the transform function and derivation of the latent values for each day of the calibration dataset, days with similar rain statistics are pooled together by rain typing. The first three principal components of the latent field are preliminarily derived from the Karhunen-Loève transform of all latent values. Next, the parameters of the GMM are inferred using an expectation-minimization approach [*Fraley and Raftery*, 2002]. Finally, rain typing (i.e., clustering) is performed by assigning to each day the type that corresponds to the most probable component of the GMM.

After rain typing, the time series of observed rain types is analyzed in relation to observations of the meteorological covariates to calibrate the non-homogeneous Markov chain. The baseline transition matrix ($\gamma_{ij}$) is first estimated by counting

the transitions between each pair of rain types occurring during the calibration period and normalizing the result by the total
number of transitions. Next, the parameters of the mean vector ($\boldsymbol{\mu}_{ij}$) and the covariance matrix ($\boldsymbol{\Sigma}_{ij}$) used to make the Markov
chain non-homogeneous are estimated by the method of moments applied to covariates observations.

Conditional to each rain type, the joint distribution of the parameters of $\psi$ is inferred by multivariate kernel density

estimation using a trivariate Gaussian kernel. The bandwidth of the kernel is selected following the Scott's rule [*Scott*, 1979]
[*Scott*, 2010], i.e., in the present case:
$$\sqrt{\boldsymbol{H}_{ii}} = N^{-\frac{1}{7}} \times \sigma_i \tag{7}$$
where $\boldsymbol{H}$ is the bandwidth matrix of the kernel, $N$ the number of days in the calibration dataset, and $\sigma_i$ the standard deviation
of the i$^{th}$ parameter (here i=1..3).

Finally, because the latent field is simulated using an analog approach (cf next sub-section for details), this part of the

model does not require formal inference.

### 2.3.3 Stochastic rainfall generation

After model calibration, stochastic rainfall generation is performed following the steps summarized in Fig. 2. Starting

from a time series of meteorological covariates, rain types are first simulated using the non-homogeneous Markov chain
described in Eq. (6). Next, conditional to this simulated rain type time series, the parameters of the transform function are
sampled from their joint distribution defined by Eq. (7). Subsequently, one realization of the latent field is simulated using an
analog approach [*Gangopadhyay and Clark*, 2005] [*Mezghani and Hingray*, 2009] [*Yiou*, 2014], i.e., by randomly picking the
latent field of a day belonging to the same rain type as the day to simulate from the calibration dataset. Finally, the simulated
rain field is obtained by back-transformation of the simulated latent field (Eq. 1–2) using the simulated parameters of the
transform function.

### 2.4 Model assessment

The ability of the model to identify climatologically relevant rain types is assessed qualitatively by applying rain

typing to the full study dataset of section 2.1 and scrutinizing the emergent spatial-temporal rainfall patterns for each type. The
resulting classification is subsequently interpreted in terms of rain generation processes by confronting rain types with co-
occurring meteorological covariates. However, in doing so, one should keep in mind that the rain typing procedure is fully
statistical and that the rain type description is based on emerging statistical patterns, not on physical modeling (e.g., using a
numerical weather model to reproduce the observed patterns).

When discussing rain types and their link to rain generation processes, special attention is paid to:

(1)  The emergence of spatial patterns in relation with orographic effects;
(2)  The seasonality of rain type occurrence in relation with the regional annual rain cycle;

(3) The relationship of rain types with the state of the atmosphere quantified by the set of climate covariates described in section 2.4 and used here at a daily resolution (i.e., not aggregated at the monthly scale as is the case for the conditioning of the non-homogeneous Markov chain).

After the assessment of the climatological realism of rain types, the ability of the model to stochastically generate rainfall is assessed using a leave-one-year-out cross-validation procedure. Data from one year are iteratively removed from the study dataset of section 2.1 and the stochastic model is calibrated using the remaining data (i.e., 19 years of data are used for model calibration). The model is fully recalibrated, which includes rain typing, inference of the transform function, and creation of a training dataset of latent fields. After model calibration, daily rainfall is simulated for each day of the target year, i.e., the year excluded from the calibration dataset. Fifty simulations are generated to assess the uncertainty associated with stochastic rainfall generation. The same procedure is repeated for each year of the study dataset, which leads to a 20-year long validation set made of 50 simulations for each gauge of the Oʻahu rain-monitoring network.

Simulated rainfall is compared to observations following a multi-criteria approach. First, simulation results are evaluated qualitatively by visual inspection of rainfall time series for the four target stations of Fig 1.a. Next, a quantitative assessment is performed using the following evaluation statistics:

(1) Site-specific rain statistics. The following statistics are considered: quantiles 10%, 50% and 90% of monthly rain accumulation to assess seasonality; annual rain accumulation to assess interannual variability; quantile-quantile (q-q) plot of the percentiles of daily rain accumulation to assess the probability distribution of daily rainfall; and q-q plot of the percentiles of wet-spell duration to assess rain persistence.

(2) Spatial patterns of rain distribution across the island. The following statistics are mapped to investigate the spatial distribution of rainfall: quantiles 10%, 30%, 50%, 70% and 90% of daily rain to assess how the probability distribution of rainfall varies in space.

(3) Areal rain statistics. Q-q plots of the percentiles of (i) the proportion of dry rain gauges, (ii) mean and (iii) max of daily rain, and (iv) the coefficient of variation of rain accumulation across the island to assess island-scale statistics.

## 3 Results

### 3.1 Rain types in Oʻahu

Figure 3 displays the 22 rain types identified for Oʻahu Island during the period 1991–2011. Although this number may seem high compared to the number of rain types inferred for mid-latitude continental climates (usually less than 10 types, see e.g. [*Vrac et al.*, 2007] [*Benoit et al.*, 2018b]), we believe that it reflects the tremendous variability of rainfall observed in the Hawaiʻi archipelago [*Giambelluca et al.*, 2013], which led Hawaiians to use more than one hundred different words to describe rainfall [*Akana and Gonzalez*, 2015]. In addition to the large number of rain types required to account for rainfall variability in tropical islands, one key attribute of the resulting classification is that although no information is given to the

classifier about geographical coordinates, time of occurrence, or meteorological covariates, the identified rain types display
well-defined patterns of spatial rain distribution (Fig. 3a), seasonality of occurrence (Fig. 3a), and correlation with the regional
state of the atmosphere (Supplementary Material 2).

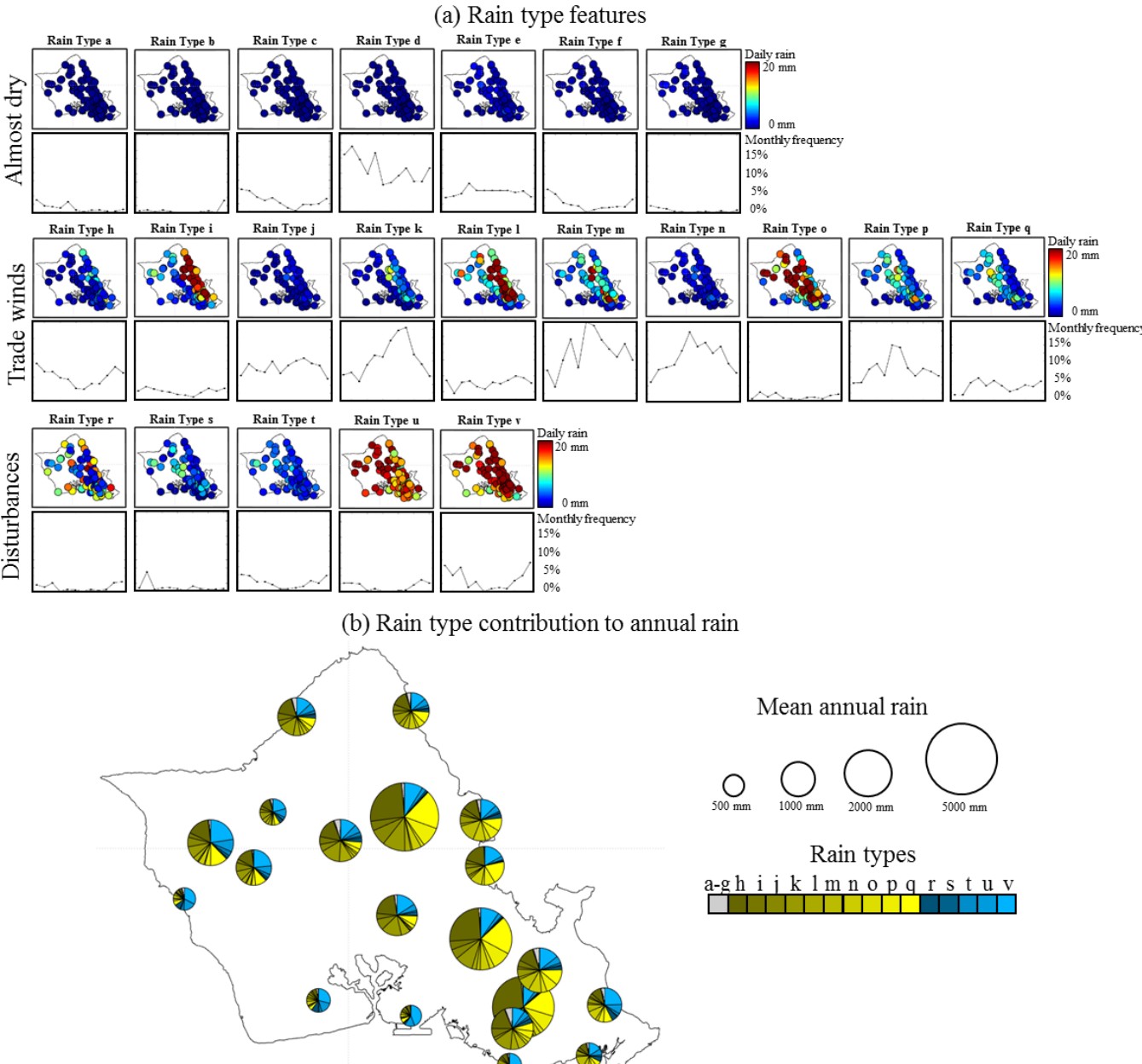


**Figure 3: Rain types identified for the island of Oʻahu.** (a) Spatial distribution of daily rain and frequency of occurrence of each rain type.
(b) Contribution of each rain type to the annual rain accumulation for a selection of 20 gauges spread throughout the island.

To better identify the main modes of rainfall variability over Oʻahu, rain types are pooled into three hyperclasses (H1-3) that can be linked to the three main rain generation processes in the area (Fig. 3):

- (H1) Almost dry days (Fig. 3, rain types a–g). During these days, most rain gauges report no rain, and no gauge reports more than 5 mm/day on average. These types of weather conditions are associated with a stable atmosphere and a low moisture flux (Fig. SM 2.1).

- (H2) Trade wind days (Fig. 3, rain types h–q). This category displays well-defined spatial patterns of rain accumulation caused by orographic lifting, and are associated with a stable atmosphere, a well-defined trade wind inversion, and an important influx of moisture below the inversion layer under the influence of east-northeasterly trade winds (Fig. SM 2.1). When scrutinizing inter-type variability within this category, note that the location of the rain maximum shifts westward with increasing moisture flux, likely due to stronger trade winds causing an overshoot of orographic rain enhancement whereby rain forms over the mountains but falls further downwind on the leeward side [*Daly et al.*, 2017]. In addition, for similar wind conditions and, therefore, spatial patterns (compare for instance types j, k and l), rain intensity is correlated to the instability of the atmosphere (Fig. SM 2.1).

- (H3) Regional atmospheric disturbance days (Fig. 3, rain types r-v). These types display either unstructured (types r–t) or relatively homogeneous (types u–v) spatial patterns of rain accumulation and are associated with low pressure, unstable atmosphere, and absent (or weak) trade wind inversion. This allows high moisture content at high altitude (Fig. SM 2.1). These rain types mostly occur during winter, i.e., the local rainy season. When scrutinizing inter-type variability within this category, note that rain intensity increases with atmospheric instability and the presence of humidity at high altitude, and that the spatial patterns tend to become more structured when the low-level moisture influx increases (probably due to stronger and more uniform winds).

Hence, rain typing provides new insights on island-scale rain climatology (Fig. 3b). In particular, this step helps us gain a better understanding of how different atmospheric conditions lead to different rain generation processes that, when interacting with island topography, generate contrasting orographic effects. In the case of the island of Oʻahu, orographic rain enhancement occurring during days influenced by trade winds is the main explanation for the high annual rain accumulations in the Koʻolau mountains (up to 5000 mm annual rainfall), while widespread rainfall linked to regional atmospheric disturbances is the main source of rain at leeward locations despite their relative temporal scarcity.

**3.2 Simulation of site-specific rainfall time series**

Figure 4 displays stochastic rainfall generation outputs for the four rain gauges of Fig 1.a which experience different rainfall climatologies. Visual inspection of the simulated time series shows that our stochastic rainfall generator is able to simulate synthetic rainfall that is almost indistinguishable from the observed one. The dry-wet ratio as well as the marginal distribution of daily rain accumulation seem properly replicated, except for the dry leeward gauge (first row in Fig. 4) where the 20-year maximum tends to be overestimated (see also Fig. 5.d). In terms of temporal variability, the simulated time series

properly capture the seasonal cycle visible for the two leeward gauges (rows 1-2 in Fig. 4), as well as the inter-annual variability
visible for the two coastal gauges (rows 1 and 4 in Fig. 4). Finally, the rainfall generator captures the relatively steady behavior
of the wet gauge located in the Koʻolau range (row 3 in Fig. 4).

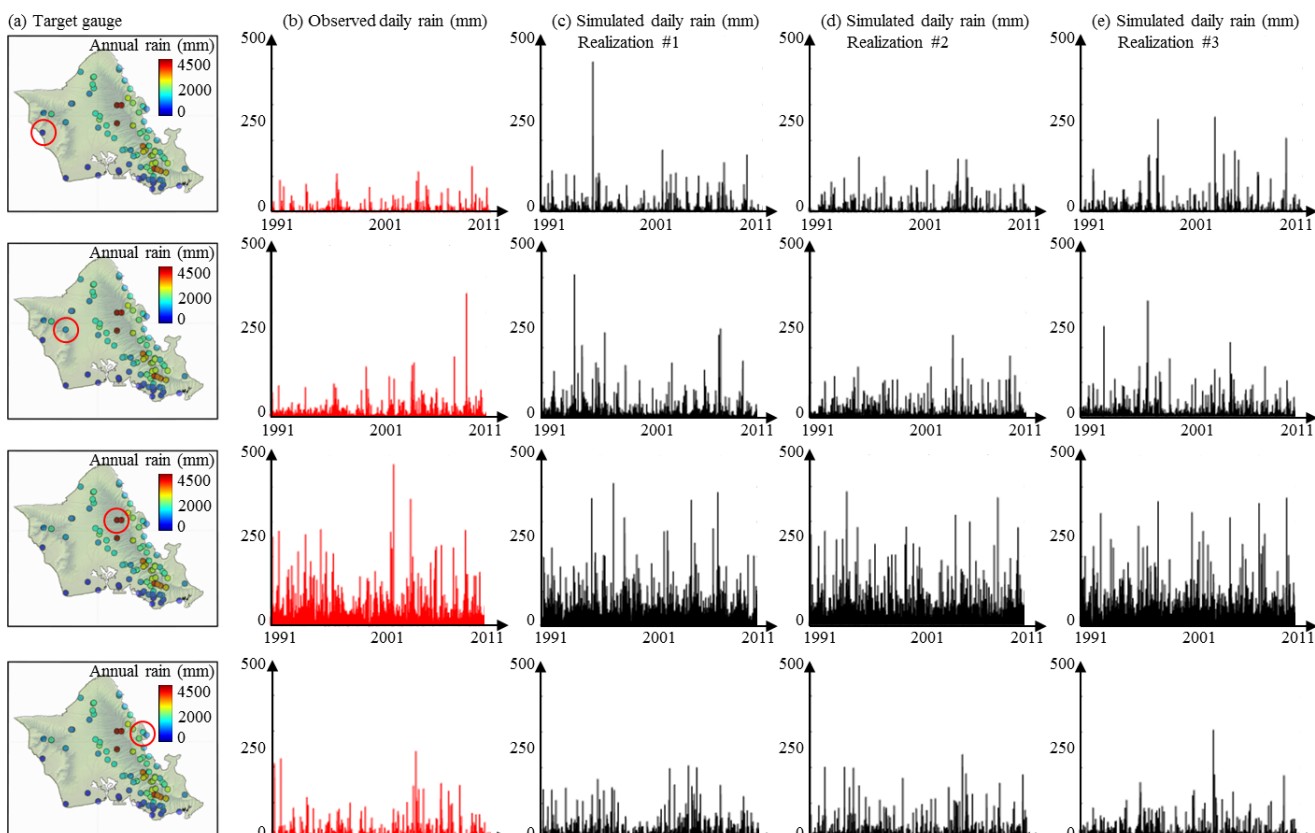


**Figure 4: Ability of the model to simulate realistic rainfall time series on Oʻahu.** (a) Target locations. (b) Observed rainfall time series
for the period 1991-2011. (c-e) Simulated time series. For readability, only the three first realizations of the 50-member ensemble are
displayed.

**3.3 Simulation of site-specific rain statistics**

To complement the qualitative assessment of site-specific rainfall time series in Figure 4, Figure 5 investigates the

statistical features of the results of the cross-validation procedure (50 realizations are drawn) for the same four rain gauges.

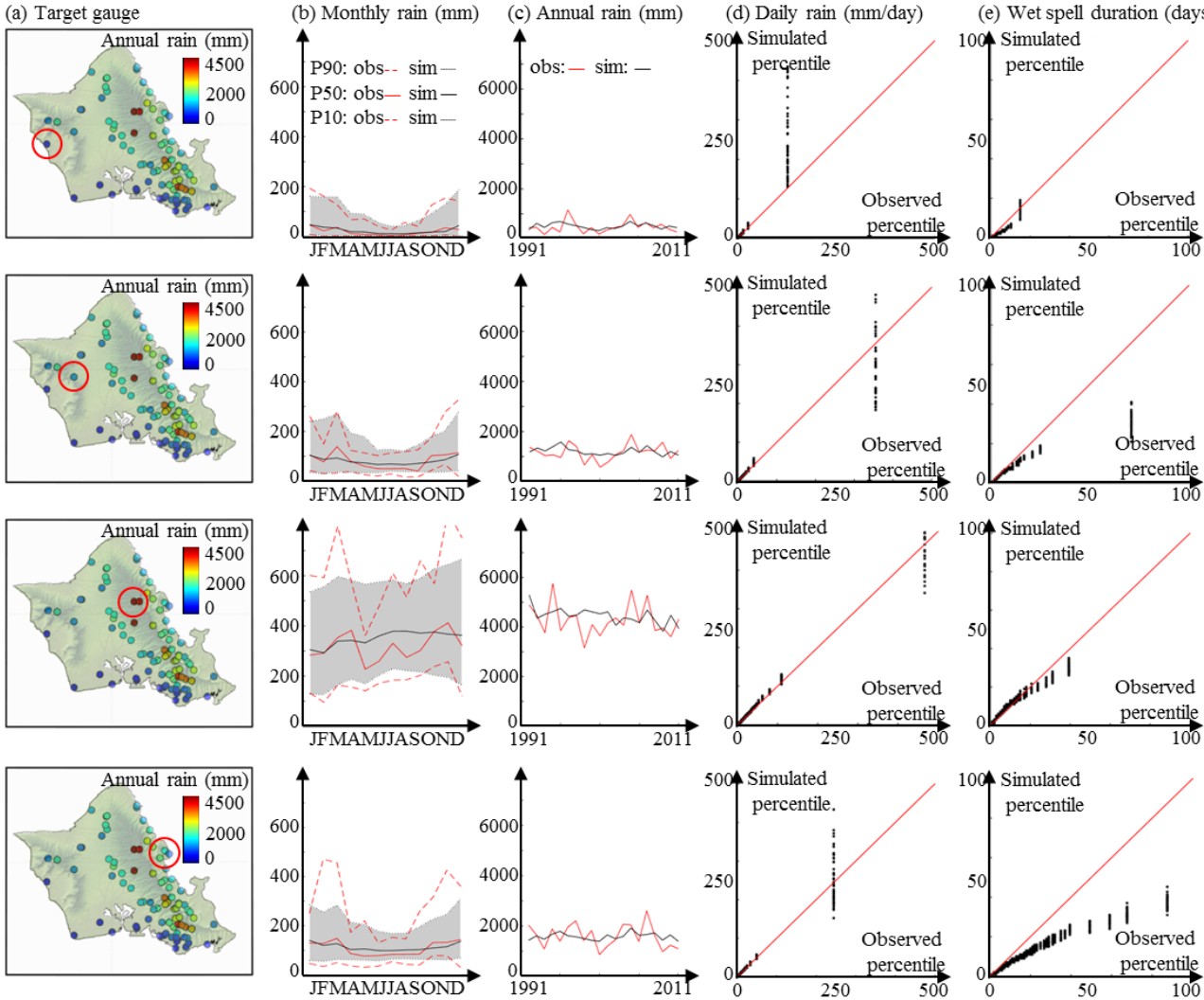


**Figure 5: Ability of the model to simulate site-specific rain statistics on Oʻahu.** (a) Target locations. (b) Observed (red) and simulated
(black) monthly rain accumulation. Dashed lines denote quantiles 10% and 90%, and solid lines denote the quantile 50%. In the case of
simulations (black), for readability, we report in the figure only the median of each quantile (10%, 50%, 90%) instead of 50 simulated
quantiles. The grey band between the dashed black lines denotes the interval Q10%-Q90% derived from simulations. (c) Observed (red) and
simulated (black) annual rain accumulation. In the case of simulations (black), for readability, we report in the figure only the median of the
50 simulations. (d) Q-q plot of daily rain percentiles. (e) Q-q plot of wet-spell duration percentiles. In (d-e) we display q-q plots between
one observation time series (abscissa) and 50 simulations (ordinates). Hence, black dots line up vertically in the q-q plots because for each
percentile, 50 simulations are compared to a single observation.

The results in Fig. 5 show that the proposed model correctly simulates rainfall seasonality (Fig. 5b) and interannual
variability (Fig. 5c). Note that simulations capture both the stronger seasonality at leeward locations (compared to windward
locations) as well as the near absence of seasonality at the wettest gauge located in Koʻolau Mountains (Fig. 5, third row). The
interannual variability of rain accumulation is also properly simulated, in particular, at leeward locations where the impact of
winter storms is the highest. These results suggest that the non-homogeneous Markov chain of order 1 conditioned to monthly-
aggregated meteorological covariates adequately models the long-term variability of rain accumulation, and that the selected
covariates properly capture rain type occurrence in a tropical marine climate.

However, rain persistence is slightly underestimated at some locations, especially for the high percentiles, i.e., long-

lasting wet spells (Fig. 5e). This result exposes limitations in the use of the non-homogeneous Markov chain of order 1 for
modeling weekly-scale temporal variability of rainfall. This may be explained by the fact that daily-scale and seasonal-scale
rainfall fluctuations are informed, respectively, by the Markov chain of order 1 and conditioning to monthly-aggregated
meteorological covariates, but that the weekly-scale is not explicitly included in the model. Although the resulting errors are
of low magnitude, they should be kept in mind in case of applications requiring a precise estimation of rain persistence, for
example crop models in semi-arid environments that can be found in some leeward areas of the target islands.

The simulations properly reproduce site-specific marginal distributions of daily rain accumulation (Fig. 5d), except

for the driest gauge (top row in Fig. 5) where the 20-year maximum tends to be overestimated. The satisfactory simulation of
rainfall distribution at several sites suggests that a type-dependent gamma distribution is an adequate model for the non-zero
daily rain accumulations across the island. It is noteworthy that all percentiles of the marginal distribution of rain accumulation
are properly reproduced in simulations, which suggests that our model is able to simulate the whole spectrum of daily
precipitation, from dry days to intense rainfall.

**3.4 Simulation of island-scale rain fields**

Figure 6 displays the results of the cross-validation procedure focusing on island-scale features. Figure 6a compares

observed and simulated spatial patterns for five quantiles of daily rain accumulation across the island of Oʻahu. Results show
very good model performance in reproducing the spatial patterns of daily rainfall. This result was expected because the use of
empirical copulas combined with rain typing is almost equivalent to resampling the observed spatial patterns conditional to
meteorological covariates. However, satisfactory simulation results ensure that the rain-type-based resampling of latent fields
is unbiased and that the choice and calibration of the meta-Gaussian model are relevant for the study island.

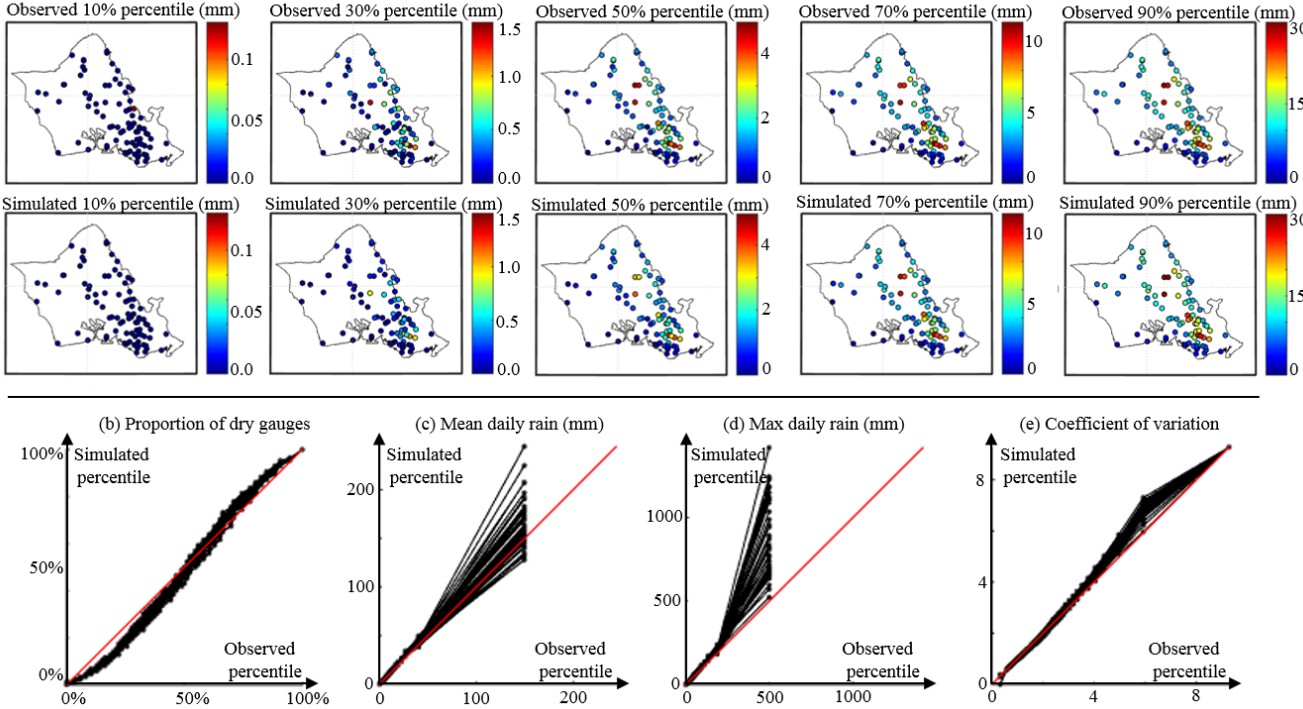

**Figure 6: Assessment of island-scale statistics simulation in Oʻahu.** (a) Spatial patterns of observed (upper row) and simulated (lower
row) percentiles of daily rain accumulation. From left to right: 10%, 30%, 50%, 70% and 90% percentiles. (b–d) Q-q plots of key rain
statistics aggregated over the whole rain gauge network: (b) proportion of dry gauges; (c–d) mean and max daily rain; (e) coefficient of
variation. Corresponding pdfs are displayed in Supplementary material 3. In (b-e) we display q-q plots between one observation time series
(abscissa) and 50 simulations (ordinates). Hence, black dots line up vertically in the q-q plots because for each percentile, 50 simulations are
compared to a single observation. To distinguish between simulations, quantiles related to the same simulation are connected by a solid
black line.


Figure 6b-e assesses the ability of the model to simulate four key rain statistics - the proportion of dry gauges, mean
and max of daily rain accumulation, and coefficient of variation of daily rain across the island - aggregated over all rain gauges
of the rain monitoring network of Oʻahu. Results show a slight underestimation of the low percentiles of the proportion of dry
gauges (Fig. 6b), which corresponds to a slight overestimation of the frequency of proportions of dry rain gauges below 5%
(i.e., 4 gauges over 86 in the present setting). That is, our model tends to simulate rain at all 86 gauges when a very small
number of gauges (less than 4) actually records no rain, leading to a slight drizzle effect in space. A careful examination of the
spatial patterns of daily rain percentiles (Fig 6a) shows that the patterns of spatial intermittency are properly simulated, which
suggests that the drizzle effect is randomly distributed amongst locations, thus reducing its potential impact for applications.
This level of accuracy in the simulation of the rain fraction shows that a truncated Gaussian latent field is an appropriate model
for rain intermittency. In addition, the correct simulation of the spatial patterns of dry locations in Fig. 6a suggests that the
distance-based modeling of the censored latent values (Eq. 2) coupled with empirical copulas is a proper model for the spatial
distribution of dry locations. Similarly, the good agreement between observed and simulated coefficients of variation (Fig. 6e)
coupled with the correct simulation of spatial patterns of non-zero daily rain accumulation in Fig. 6a suggest that the selected
meta-Gaussian framework captures the spatial distribution of non-zero rain accumulations.

Finally, Fig. 6c-d shows that island-scale daily mean and maximum rain accumulation are properly simulated, despite

an overestimation of the last percentile of the maximum, i.e., the 20-year maximum observed over the whole island. This result
suggests that the meta-Gaussian framework coupled with the kernel estimation of the transform function parameters performs
reasonably well to reproduce the marginal distribution of island-scale rain accumulation. However, the attempt to reproduce
both island scale statistics and site-specific marginal distributions (from dry days to heavy rainfall) results in an inaccurate
simulation of the island-scale 20-year extreme precipitation. This limitation calls for additional developments before the
proposed model can be used for simulating extremes in a spatial context [*Opitz et al.*, 2021].
**3.5 Model versatility**

To investigate the versatility of our stochastic daily rainfall model, the case study performed in sections 3.1–3.3 for

the island of Oʻahu (Hawaiʻi, USA) located in the North Pacific was repeated in supplementary material 4 for the island of
Tahiti (French Polynesia) located in the South Pacific. Figure 7 provides a brief overview of the results. This additional cross-
validation shows that our model also performs very well for Tahiti, despite a wetter (annual rain reaches 10 000 mm in Tahiti)
and more seasonal climate than the Oʻahu case study. In addition, the model adapts automatically to different dataset sizes (86
rain gauges x 21 years for Oʻahu, 26 gauges x 11 years for Tahiti) and rain climatologies due to the selection of different
numbers of rain types. These results suggest that our model may be adapted to most high tropical islands across the globe.

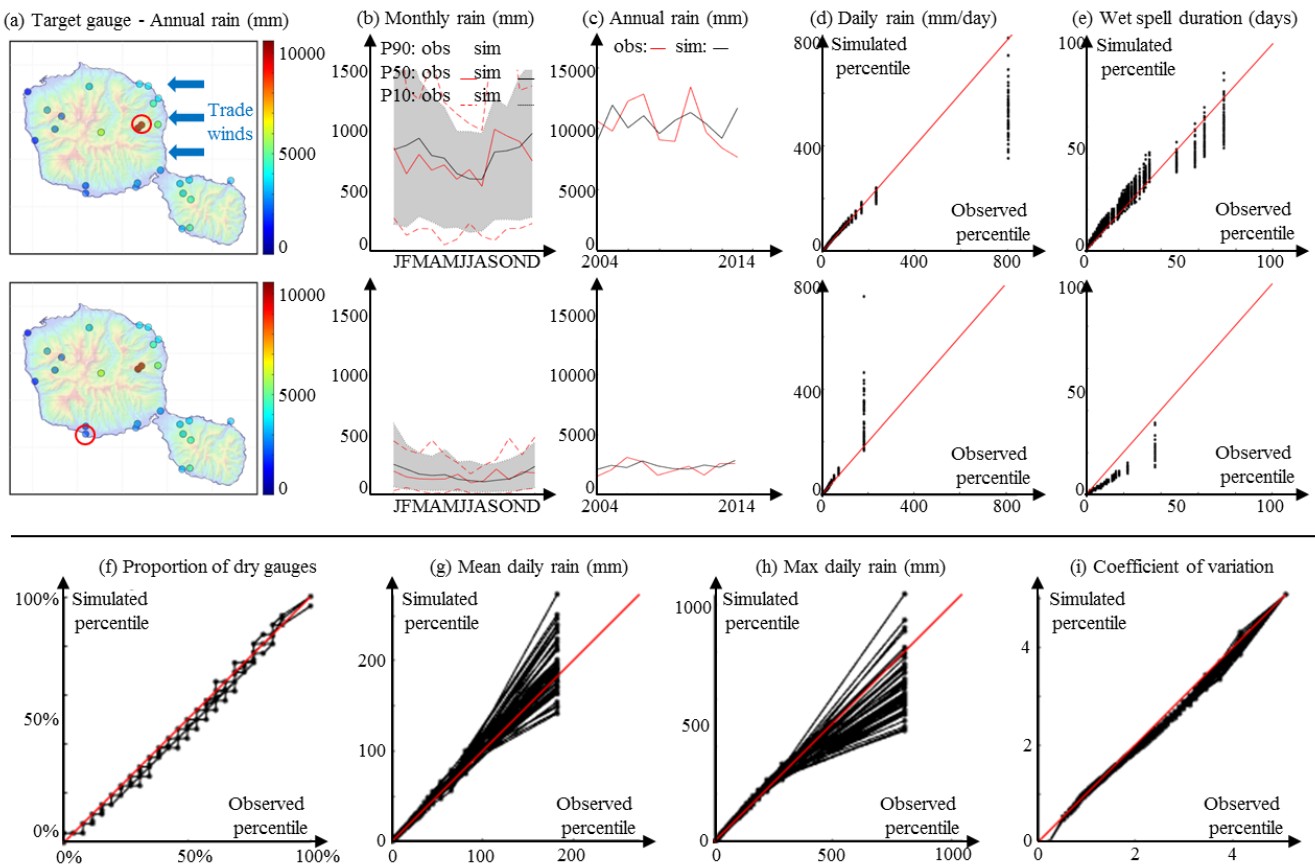

**Figure 7: Overview of stochastic daily rainfall simulation for the island of Tahiti, French Polynesia.** (a-e) Site-specific statistics: (a) Target locations. (b) Observed (red) and simulated (black) monthly rain accumulation. (c) Observed (red) and simulated (black) annual rain accumulation. (d) Q-q plot of daily rain percentiles. (e) Q-q plot of wet-spell duration percentiles. (f-i) Island-scale statistics: (f) proportion of dry gauges; (g–h) mean and max daily rain; (i) coefficient of variation.

## 4 Discussion and conclusion

### 4.1 Discussion: stochastic modeling of orographic rainfall patterns

Validation results in section 3 show that the proposed model is able to simulate realistic multi-site rainfall time series, which accurately reproduce site-specific and island-scale daily rain statistics for two different tropical islands. This has been made possible by a two-step modeling approach (rain typing and meta-Gaussian representation of island-scale daily rainfall), which takes into account our conceptual knowledge about orographic rain enhancement in tropical islands.

The first component consists of rain types, which summarize island-scale rain statistics. Unlike weather type based approaches [*Ailliot et al.*, 2015] [*Réchou et al.*, 2019], we define rain types based on rain features only, i.e., no information

about meteorological covariates or large-scale circulation are included during the classification step. This leads to a classification centered on rainfall intensity and spatial distribution, which allows us to explore how island-scale rainfall variability is impacted by orographic effects (section 3.1). The links between rain types and local climate are established in a second step by conditioning the non-homogeneous Markov model of rain type occurrence to meteorological covariates. We conceptualize rain types as the main modes of island-scale daily rainfall variability, which is assumed to be primarily influenced by orographic effects caused by interactions between changing atmospheric conditions and fixed island topography. In this context, one interesting contribution of this study is the refinement of the meteorological predictors proposed by [*Sanfilippo*, 2020] for rain type occurrence in a tropical marine climate, in particular, to distinguish between shallow convection occurring during typical trade wind situations and deeper convection in the vicinity of atmospheric disturbances.

The second component of the model consists of a meta-Gaussian representation of island-scale daily rainfall. By explicitly separating rain intensity and spatial distribution, this representation contributed to the performance of the rain typing procedure detailed above and in the identification of rain types with well-defined spatial patterns. When used for stochastic rainfall generation, the adopted meta-Gaussian representation performed well in simulating site-specific rain statistics as well as island-scale spatial patterns of daily rain accumulation. This good performance can be explained by two factors. First, the determination of the censored latent values based on the distance to the closest wet gauge (Eq. 2) generates realistic spatial patterns of dry areas and dry-wet transition [*Schleiss et al.*, 2014]. This contributes to the proper modeling of the spatial intermittency of daily rain fields in tropical islands, which is caused by the drying effect of sinking air masses after crossing mountains. The second innovation of the model is the joint use of empirical copulas and a parametric transform function to model the spatial patterns of non-zero rain. It has the advantage of faithfully preserving the spatial rainfall patterns while generating unobserved values through the kernel density estimation of the transform function parameters distribution. The choice of mimicking the observed spatial rainfall patterns as closely as possible is justified by the complexity of orographic effects and associated rain gradients in tropical islands [*Giambelluca et al.*, 2013] [*Laurent et al.*, 2019] [*Benoit et al.*, 2021].

**4.2 Concluding remarks**

In this paper we presented a new stochastic daily rainfall generator dedicated to high tropical islands. The combination of (i) rain types, (ii) a non-homogeneous Markov model of rain type occurrence conditioned to meteorological covariates, and (iii) a meta-Gaussian representation of the spatial distribution of daily rainfall allowed us to generate realistic daily rain fields honoring both site-specific and island-scale rain statistics. The performance of the model was carefully tested and illustrated for the islands of Oʻahu (Hawaiʻi, USA) and Tahiti (French Polynesia), both located in the tropical Pacific. Cross-validation results prove the ability of the model to capture and simulate the main features of daily rainfall over these two high tropical islands.

The main strength of our model is its ability to simulate diverse patterns of multi-site daily rainfall, as well as their linkage with regional atmospheric conditions. It represents a new tool for stochastic investigation and modeling of orographic rain enhancement on tropical islands with complex topography. The main limitations, however, are (i) the inaccurate

simulation of extreme rainfalls, which calls for caution when using our model for flood risk assessment, and (ii) the restriction
to multi-site simulation, which calls for an additional step of stochastic interpolation of the multi-site patterns when gridded
outputs are required.

Because of the above strengths and limitations, the main envisioned applications relate to impact studies that require
detailed knowledge of daily precipitation in tropical islands, in particular, when the spatial pattern of rainfall plays an important
role. This includes watershed water resources management and eco-hydrological studies.

Our model can also be used for the stochastic downscaling of future precipitation projections and thereby contribute
to the current efforts to better understand, manage, and secure tropical island water resources in a changing climate. An
essential future investigation in this direction will be to assess how well GCMs simulate the set of meteorological covariates
we selected to drive our stochastic rainfall generator, in particular the vertical temperature gradient used to inform shallow
convection.


*Code and data availability.*
The implementation of the proposed stochastic rainfall model is open source (MATLAB implementation) and freely available
in the following repository (https://github.com/LionelBenoit/StochasticRainfallGenerator_TropicalIslands). The dataset of
daily rainfall observations on Oʻahu is open data and freely available on the Hawaiʻi Climate Data Portal
(https://www.hawaii.edu/climate-data-portal/data-portal/). An extract of this dataset is available in MATLAB format as a code
demo in the same repository as the source code of the model. The dataset of daily rainfall observations on Tahiti is available
upon request from Météo France (contact.polynesie-francaise@meteo.fr) and Groupement d'Etudes et de Gestion du Domaine
Public de Polynésie Française (secretariat@equipement.gov.pf).
*Author contributions.*
LB, LS and TWG designed the experiment. MPL and LS compiled the daily rainfall datasets of Oʻahu and Tahiti respectively.
LB and ADN selected the meteorological covariates and designed the non-homogeneous Markov Chain. LB and MPL designed
the meta-Gaussian model and the rain typing method. LB implemented the model and performed the numerical experiments.
LB wrote the paper with input and corrections from all co-authors.

*Acknowledgments.*

The work of Lionel Benoit is funded by the Swiss National Science Foundation (SNSF), grant number P2LAP2_191395. The work of Lydie Sichoix is supported by the Government of French Polynesia - Ministère de la Recherche through the project E-CRQEST, grant number 05832 MED 08/26/2019. The authors are grateful to the Hawai'i Climate Data Portal for providing the daily rainfall dataset of the island of O'ahu, and to the French Weather Agency (Direction Interrégionale en Polynésie française - Météo France) and the Polynesian public service named Direction de l'Equipement (Groupement d'Etudes et de Gestion du Domaine Public de Polynésie Française - GEGDP) for providing the daily rainfall dataset of the island of Tahiti. The authors are grateful to May Izumy from the Publication services of the School of Ocean and Earth Science and Technology, University of Hawai'i for proofreading this manuscript.

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
