# Peer review of "Stochastic daily rainfall generation on tropical islands with complex"

_Hydrology and Earth System Sciences, 2021_

## Referee Comment (RC2)

[revised manuscript text omitted]

I think that this model will not be confined to tropical islands. I can see applications in my country, South Africa, where we have high coastal mountains in the East rising to 3000m flattening through dryer areas towards the North West. Check the figure from our work ; precip. max of 2000mm: "The gridded Mean Annual Precipitation averaged over each of the 1946 quaternary catchments in the region" in: Pegram GGS, Scott Sinclair and András Bárdossy (2016). New methods of infilling Southern African raingauge records enhanced by Annual, Monthly and Daily Precipitation estimates tagged with uncertainty. *Water Research Commission,* WRC Report No. 2241/1/15 ISBN

[revised manuscript text omitted]

---

## Author Comment (AC1)

Response to the comments of Reviewer#1

*Reviewer comment (RC 1.1): Some of the ideas in the paper are potentially interesting. However, the existing literature on weather generators (particularly for daily precipitation) is vast: to justify yet another new approach therefore, it is necessary to demonstrate that it improves on existing methods in some way (it could be the performance of the method, its ease of implementation, its range of applicability, its computational feasibility etc.). The authors do not provide any such demonstration, perhaps because they don't seem aware of the state of the art in the area.*

Authors' response (AR 1.1): In our opinion there are two main novelties in our study, which we briefly summarize hereafter and will discuss in detail throughout the present response to Reviewer#1 comments:

- Our model has been designed to account explicitly for rainfall patterns, i.e., not only inter-gauges correlation, as is often the case in multi-site stochastic weather generators (SWGs) (e.g., *[Mehrotra and Sharma, 2007]* [*Bennett et al.*, 2018]), but also rain location within a given target area (here a tropical island). In the present context, this allows us to infer which part of the island receives the most rainfall for a given time of the year, which is a very important feature of the climate of interest. It is indeed important to notice here that in tropical islands, rain location changes drastically with the course of the year (see Fig. 3a and Fig. SM2.1 for examples of spatial rainfall patterns in Hawai'i and French Polynesia respectively, and Fig. 4 and SM2.4 to see how these patterns influence site-specific rain seasonality).
- Our model accounts for the rain climatology of tropical islands. As stated in our introduction, we believe that stochastic rainfall generators (SRGs) should be tailored to accommodate the climate at hand, which is an avenue followed by several other researchers in the past (e.g., [*Peleg and Morin*, 2014] [*Wilcox et al.*, 2021]) and has led to valuable developments. We agree with Reviewer#1 that generic methodological developments are the foundation of SRGs, but we would add that the adaptation of pre-existing modelling blocks to different climates is a necessary extension of this theoretical effort. The present study follows this perspective, and tries to take advantage of the combination of the (pre-existing) concepts of rain typing (e.g., [*Oriani et al.*, 2017] [*Benoit et al.*, 2018a]), transformed Gaussian random fields (e.g., [*Baxevani and Lennartsson*, 2015] [*Allard and Bourotte*, 2015]) and analogs resampling (e.g., [*Gangopadhyay and Clark*, 2005] [*Yiou*, 2014]) to design a SRG adapted to the specific case of tropical islands. To the best of our knowledge, this topic has not yet been adequately addressed.

To emphasize this second point (the development of a new model dedicated to tropical islands), which was the motivation for our study, we decided to focus our introduction on the adaptation of SRGs to the climate of the area of interest. Unfortunately, this may have given the false impression that we neglected the many alternative modelling choices existing in the literature (e.g., Hidden Markov Models, Generalized Linear Models, and object based models). This false impression may have been reinforced by our initial choice to focus our discussion on the justification of our modelling choices instead of discussing how they fit within the broad literature of SWGs. We see now that it is not the best choice, and will improve our paper in this respect by substantially expanding the discussion section to include the points from this response.

*RC 1.2: Literature that seems particularly relevant includes that on Hidden Markov Models (e.g. Hughes et al. 1999; Ailliot et al. 2009): this uses the same basic idea of classifying each day on the basis of the joint spatial distribution of precipitation, but does so in what seems to be a more principled way than the present paper.*

AR 1.2: Hidden Markov Models are indeed broadly used in the SWGs literature, and we agree that they rely on the same idea as our method of defining rain types, namely the classification of each day based on rainfall (or more generally weather) statistics. The only difference we see between HMM and our method (where rain types transition according to a non-homogeneous Markov Chain) is that in one case the states of the Markov Chain are hidden (HMM), and in the other case they are explicit (the present approach). This second option is not new in the field of SWGs (see [*Ailliot et al.*, 2015] for a review of SWGs based on weather types, discussing both hidden and directly observed types), and has been shown to be valuable in several past studies (e.g., [*Bárdossy and Plate*, 1991] [*Bárdossy and Plate*, 1992] [*Wilby*, 1994] [*Vrac et al.*, 2007a], to cite but a few). We chose the explicit approach over the HMM in the present study because several steps of our modelling chain rely on non-parametric methods, which prevents the formulation of the likelihood of the full model and in turn the fit of an HMM.

*RC 1.3: Moreover, approaches using generalised linear models with topographical indices as covariates (e.g. Ambrosino et al. 2014; Chandler 2020) address the issue of topographical variability directly: it's not obvious to me that such approaches would fail in a tropical island setting.*

AR 1.3: We agree that Generalized Linear Models (GLMs) can address the issue of topographical variability in an elegant way in case of moderate and relatively steady orographic effects, but we see several obstacles to the application of GLMs in the present case where orographic rain enhancement is very strong and variable in space and time.
First, rain accumulation in tropical islands is not a simple function of altitude, but is strongly modulated by the exposure of slopes to prevailing winds. This aspect, common to temperate and tropical mountains, can be accounted for within the GLM framework, but requires the inclusion of information about slope orientation at a scale that properly represents rain-topography interactions (e.g., [*Chandler*, 2020]), which is not always easy to define in case of complex topography (e.g., [*Daly et al.*, 2017] [*Foresti and Pozdnoukhov*, 2012]) as encountered in tropical islands. This situation is further complicated by the possibility of successive mountain chains along the track of the moisture flux, with the consequence that the second chain is drier than the first one. This phenomenon can be seen on Oʻahu, where the summit of the Waianae range (Mt Kaʻala, 1227m, 2015 mm/year) is significantly drier than the summit of the Koʻolau range (Puʻu Kōnāhuanui, 960m, around 3000 mm/year). A possible predictor for this can be derived from the integral of positive height difference along the moisture flow, but since this flow is not constant through time it would involve complex pre-calculations. In addition, many local effects add to this already complex situation, in particular the presence of rain enhancement overshoot in many valleys (rain maximum is slightly leeward from the main crest, see e.g., [*Giambelluca et al.*, 2013] [*Benoit et al.*, 2021]), and the probable influence of the angle between the moisture flux and the mountain ridge, which makes the Northern Koʻolau (max annual rain >6000mm/year) significantly wetter than the Southern section of the range (max annual rain ~4000mm/year). These local effects are still only partially understood, which makes the selection of predictors and the setting of GLMs particularly difficult in such areas. Finally, the above features are drastically modulated by atmospheric conditions, which would require the inclusion of atmospheric covariates into the GLM, and most importantly the proper parametrization of their interaction with orographic effects. This has been shown to be possible in simple settings (as is the case for the

references proposed by Reviewer#1: [*Ambrosino et al.*, 2014] [*Chandler*, 2020]), but we are not aware of the application of such a framework in areas with complex patterns of orographic rain enhancement such as the ones described above, where a parsimonious parametrization of GLMs seems very challenging to obtain.

All in all, using GLMs in the context of tropical islands may be possible, but would certainly require a careful selection of atmospheric and topographic predictors, as well as the parametrization of their interactions and links with site-specific rainfall. This seemed like a very cumbersome process to us (partly because none of us is an expert in the field of GLMs), and as an alternative we chose a resampling approach focused on island scale rainfall patterns. Our approach has the advantage of directly and explicitly modelling: (1) rain location within the island, (2) island-scale rainfall pattern, and (3) the link between (1) and (2) and regional atmospheric conditions. This choice has been guided by a preliminary analysis of both island-scale and site-specific rain statistics, which has revealed the complex patterns of orographic rain enhancement we detailed above.

*RC 1.4: I am also surprised that the paper doesn't cite Maraun et al. (2010) which has become almost the canonical reference for anyone working in this area.*

AR 1.4: Maraun et al. (2010) is indeed a canonical reference in the field of climate projections downscaling, with a section dedicated to stochastic weather generators in this specific context. However, precipitation downscaling under climate change is only one possible application of our model, which can also be used for instance to study rainfall variability or to carry out impact studies under current climate. That is why we initially chose references more directly linked to stochastic weather generation [*Richardson*, 1981] [*Wilks and Wilby*, 1999] [*Ailliot et al.*, 2015], which we found more topical. But we agree that precipitation downscaling is a major application of our model, and we will add the proposed reference in the revised manuscript.

*RC 1.5: In view of the concerns above, as well as some technical issues (see detailed comments below), I don't think the paper merits publication in its current form. To make the case, the authors need to demonstrate that their approach improves on existing methods in some way as described above. Ideally, this would be done by carrying out an informed and fair comparison with a leading alternative method: if this isn't possible then the authors should explain why, and should offer some informed discussion of how their approach might reasonably be expected to compare.*

AR 1.5: As mentioned above (AR 1.1 to AR 1.3), we will address this comment by discussing in detail our modelling choices and by incorporating the related clarifications in the discussion section. This will allow for a better contextualization of our model within the broad literature of SWGs / SRGs.

*RC 1.6: Line 31: this dismissal of "detailed physical modelling of rain generation processes" seems quite one-sided and poorly informed. It is true that stochastic models are computationally faster than physical ones, but physical models do have their own advantages which are not acknowledged here.*

AR 1.6: We agree that our introduction of physical models was simplistic, which can be misleading. We will expand this description, and better acknowledge the pros and cons of both physical and stochastic models. This will include (but not be restricted to) the discussion of the reference [*Maraun et al.*, 2010] proposed by Reviewer#1.

*RC 1.7: Line 79: what proportion of the data have been "gap-filled"? How is the gap-filling distributed across time and between stations? How was the filling done?*

AR 1.7: Only stations with less than 5% gaps in raw data have been kept for processing. Gaps are usually spread across the whole duration of the dataset. The gap-filling has been performed using the normal ratio method [*Paulhus and Kohler*, 1952] for the Oʻahu dataset (in the main text), and using the vector sampling approach [*Oriani et al.*, 2020] for the Tahiti dataset (in supplementary materials).

*RC 1.8: I will add that gap-filling in highly variable situations is, in my view, difficult and potentially dangerous because it will tend to underestimate variability. In my view therefore, for a stochastic rainfall model to be suitable for widespread use, it must be capable of handling incomplete datasets.*

AR 1.8: We agree that incomplete datasets must be easily handled, that is why we selected (and implemented in our software) a pattern-preserving gap-filling method [*Oriani et al.*, 2020] for cases where no case-specific gap-filling has been performed (e.g., the Tahiti dataset, see supplementary material 2). This method has the advantage of being compatible with our modelling framework focusing on rainfall patterns. When a case-specific gap-filling has already been applied and validated (as is the case for the Oʻahu dataset [*Longman et al.*, 2018]), we chose to base our study on gap-filled datasets.

*RC 1.9: (in particular, what would you do if you needed to generate precipitation at a site for which you have no data? This requirement is common in many realistic applications). Lines 216-217 suggest that the proposed methodology cannot handle incomplete datasets: this is a serious limitation that needs to be acknowledged clearly and openly.*

AR 1.9: As suggested in lines 216-217 and with the terminology "site-specific" used throughout the paper, our model is indeed designed to simulate rainfall only at sites where data are available. It therefore belongs to the family of multi-sites stochastic rainfall generators (e.g., [*Breinl et al.*, 2017]) rather than the one of rain field generators (e.g., [*Paschalis et al.*, 2014]). In case precipitation needs to be generated at an ungauged location (i.e., where no data is available), we would first perform multi-site rainfall generation using the present model, and in a second step perform stochastic interpolation of these simulated values using a model that withstand non-stationarities in both space and time (e.g., [*Benoit et al.*, 2018a] [*Benoit et al.*, 2021]). However, the way uncertainties would propagate along the simulation chain remains to be investigated, and we therefore prefer to keep this interpolation step out of the scope of the present study. But Reviewer#1 is right to mention that it is a limitation of our model, and we will acknowledge it explicitly in the revised manuscript, in particular when summarizing the limitations of the model in the conclusion (lines 423-426 of the current numbering). The possible coupling with stochastic interpolation will be mentioned as well.

*RC 1.10: Lines 101-105: although the spatial variation in precipitation statistics initially seems dramatic here, a more considered inspection reveals that the statistics are more or less constant along NW-SE transects and that the predominant variation is basically along the axis of the trade winds. If one were to scale the seasonal cycles at the specimen locations to a common scale (e.g. proportion of annual rainfall falling in each month), I doubt whether they would be dramatically different. I don't see*

*anything here that really challenges state-of-the-art multisite weather generators, therefore – and hence nothing that really necessitates the development of a new modelling framework.*

AR 1.10: We disagree with this statement. Although an inspection of rain statistics limited to the annual scale (i.e., based on Fig. 1a central panel only) can indeed give the impression that the predominant variation in precipitation is fully driven by trade winds, we would like to emphasize that a more careful inspection of rainfall features at the monthly to daily scale leads to very different results (see e.g., Fig. 3a and Fig. 4b). This is the motivation for section 3.1, which aims to show that spatial rainfall patterns drastically change with seasons and atmospheric conditions (this is even more obvious in supplementary material 2.2 and Fig. SM2.1, which focus on the more seasonal climate of Tahiti).
If one looks at the seasonal cycle at specimen locations as proposed by reviewer#1 (and illustrated in Fig 4b and Fig SM2.3), one can observe that seasonality actually differs between locations. For instance, in Koʻolau mountains, one can observe a weak seasonality with three distinct local maxima in April, November and July (this last one occurring during the dry season at the island scale). In contrast, at leeward stations, one can observe a single and marked rain maximum during the NDJFM (i.e., wet season). This difference of seasonality at specimen locations is well captured by our model (Fig 4b), and this good performance can be attributed to the careful modeling of the links existing between spatial rainfall patterns and atmospheric conditions in our model.

*RC 1.11: Line 133: what's the justification for the formulation in equation (1)? It seems a bit ad hoc. Also, as defined, $Z_i$ isn't latent because it's directly connected to observable quantities.*

AR 1.11: Equation 1 simply means that we assume that a Gamma function is able to transform the observed rainfall (R) to a censored Gaussian variable (Z), such that when rain is observed the variable Z takes a value above a given threshold ($\Phi^{-1}\left(\frac{N_d}{N_t}\right)$), and when no rain is observed the variable Z takes a censored value below this threshold (the way we assign this censored value is given by Eq. (2)). The use of a parametric transform function coupled with censoring to relate observed intermittent rainfall to a latent Gaussian variable is common in stochastic rainfall models based on Gaussian processes (see e.g., [*Glasbey and Nevison*, 1997] [*Allcroft and Glasbey*, 2003] [*Allard and Bourotte*, 2015] [*Baxevani and Lennartsson*, 2015] to cite but a few). The choice of the Gamma function as a transform function may seem arbitrary, but has been selected here for the flexibility of this transform function to accommodate day-to-day variations of the distribution of non-zero rain accumulation across the island. In addition, it should be noticed that this choice is not disconnected from the literature on this topic, since a Gamma transform has already been used e.g., by [*Kleiber et al.*, 2012] in a different climate (Argentinian pampa).
Regarding the terminology, the term 'latent' is used here to describe the variable Z because Z is not directly observed. This is a common terminology in the literature related to stochastic rainfall models based on Gaussian processes (e.g., [*Glasbey and Nevison*, 1997] [*Allcroft and Glasbey*, 2003] [*Allard and Bourotte*, 2015] [*Baxevani and Lennartsson*, 2015]).

*RC 1.12: Line 142: similarly, what's the justification for the distance-weighting in equation (2)? This again seems a bit arbitrary, although it's certainly true that what is sometimes called the "spatial intermittence" problem is hard to resolve satisfactorily. Nonetheless, in my view it's not clear that the authors' proposal improves on that by, say, Stehlík and Bárdossy (2002) – another example of literature that they seem unaware of, or at least haven't considered as carefully as they should have done. And once again, it's not a "latent" field if it's defined in terms of interpretable quantities.*

AR 1.12: Equation 2 is indeed an attempt to deal with spatial intermittence. The idea behind our choice of distance-weighting is that a location far from any wet gauge should remain dry even after combining the copula at hand with slightly different parameters of the transform function during the simulation step. It is therefore application-specific and tightly linked to our other modelling choices. We agree with reviewer#1 that spatial (or temporal) intermittence is a problem that is hard to solve in a generic way in the framework of a transformed and censored Gaussian process. A large spectrum of methods have been proposed to solve it in case-specific applications. On one end of this spectrum, the option is to consider non-censored values as the main source of information about the whole process (including censored values). The spatial intermittence problem is then solved by simulating many times the full process conditional to non-censored values only, and selecting the simulations leading to patterns of spatial (and/or temporal) intermittency similar to the observed ones (e.g., [*Allard and Bourotte*, 2015] [*Papalexiou et al.*, 2018]). On the opposite end of the spectrum, more complex methods have been proposed to simulate the censored values conditional to (1) non-censored neighboring observations, (2) space (and/or time) patterns of rain intermittence, and (3) the fact that the latent variable is censored at the location of interest (e.g., [*Bárdossy and Pegram*, 2016] [*Benoit et al.*, 2018b]). Here we place ourselves half way between these two approaches, and we regard the latent variable as a rain potential that is decreasing when observed daily accumulation decreases (Eq. 1), reaches $\Phi^{-1}\left(\frac{N_d}{N_t}\right)$ at the wet/dry transition (Eq. 1 and 2), and keeps decreasing when the distance to wet gauges increases (Eq. 2).

*RC 1.13: Lines 149-150: there's a claim here that the complex spatial distribution "prevents the use of a simple parametric form ... for the spatial copulas". It's not entirely clear to me what this means, but I assume it's something like: you can't find a "standard" spatial covariance model to represent the dependence structure between the $Z_i$. I'm prepared to believe this (although it could be partly an artefact of the artificial and deterministic distance-dependence structure of equation (2)), but it would be helpful to see some plots to justify it.*

AR 1.13: Yes, we mean that the joint distribution of copulas between sites does not follow a stationary multi-variate Gaussian distribution, which implies that we cannot find easily a "standard" spatial covariance model to represent the dependence structure between the Zi. This is illustrated in the figure below, which displays bivariate scatterplots of the latent values at the 4 target stations used in Fig 1 and 4 for illustration purposes, and which shows that bi-variate spatial dependences are neither bi-variate Gaussian (i.e., scatterplots do not have an elliptical shape) nor stationary (i.e., scatterplots are centered on different means, and have different spreads). We will add this figure to the supplementary material section and discuss it to better justify our choice of non-parametric copulas to simulate the spatial dependence structure.

[Figure]

RC 1.14: Lines 158-159: the ability to calculate principal components of the "latent" field shows that it isn't latent. See earlier comments on this.

AR 1.14: Please refer to our response AR 1.11 above.

RC 1.15: Sec 2.2.3: I find this "rain typing" approach, which seems to be derived from techniques that are popular in the machine learning community, to be rather clumsy compared with the much more principled approach taken by other authors using HMMs. Even if one were to accept that the approach is worth considering, the classification in lines 172-174 is inappropriate because it fails to account for the uncertainty in each day's weather type (e.g. if you had three states then you could find that the probabilities are always (0.34, 0.33, 0.33) in which case you would assign every day to state 1 which is clearly nonsense – exactly like the electoral system in some recently failed democracies).

AR 1.15: As mentioned in AR 1.2, we adopted 'explicit' rain types (i.e., not hidden) to enable the non-parametric steps of our framework, which are justified by the complexity of rainfall patterns in tropical islands. This choice of explicit rain types requires in turn finding an alternative to HMM for rain typing, and for this study we chose a Gaussian Mixture Model (GMM) classification. This classification approach is widely used for clustering (e.g., [*Fraley and Raftery*, 2002]), and has been applied successfully in the context of weather and rain typing (e.g., [*Vrac et al.*, 2007b] [*Benoit et al.*, 2018a]). Regarding the classification (lines 172-174) we do not see anything wrong in our equations 4 and 5, which follow the usual GMM classification framework (e.g., [*Fraley and Raftery*, 2002]). We are not sure to understand the concern of reviewer#1. If we imagine that the probabilities of the three states (rain types) are 0.34, 0.33, 0.33 for all values of the feature-space, it means that the (multivariate Gaussian, in our case 6-dimentional) distributions of these three types are almost identical, in which case the model will automatically detect that only one single type exists ($N_C = 1$). And therefore this problematic case of rain type non-identifiability will not occur.

*RC 1.16: Line 179: there seems to be something wrong with this equation. If I understand correctly, it is something like a kernel estimate of the transition probabilities based on the Mahalanobis distance in the space of meteorological covariates; but if so, then summing the right-hand side over j should give 1 for any value of i or $MC_d$ (it's supposed to define a conditional probability distribution). It doesn't, at least if $\gamma_{ij}$ is a "baseline" transition probability (which isn't stated here, but seems to be the case based on a later description – if that's the case, then it's true that $\sum_j \gamma_{ij} = 1$ for all i but the exponential factor in line 179 messes things up).*

AR 1.16: Thanks a lot for spotting this typo in equation 6, the sign '=' must indeed be replaced by '∝'. By doing so we obtain the usual formulation of a non-homogeneous Markov Chain with transition probabilities based on the Mahalanobis distance in the space of meteorological covariates (.e.g., [*Hughes and Guttorp*, 1999] [*Vrac et al.*, 2007a]). We will correct this error in the revised manuscript. Regarding $\gamma_{ij}$ it is indeed a baseline transition probability, as mentioned l 182.

*RC 1.17: Line 184: how does the conditioning inform seasonality? This is another slightly dangerous assertion: relationships between precipitation and meteorological covariates can themselves be seasonally varying, for good physical reasons (think precipitation and temperature in temperate latitudes - they are positively associated in winter but negatively associated in summer).*

AR 1.17: The conditioning actually informs the state of the atmosphere, which is strongly correlated with rainfall features at the island scale, as shown in Fig SM1.1 and SM2.2. At the same time, the state of the atmosphere defined from a set of meteorological covariates is known to have a strong seasonal signal (e.g., [*Vrac et al.*, 2014]). Hence, the link with seasonality is indirect, but is still very efficient in the present case to encode seasonal and inter-annual variations of rain accumulation at specimen locations as shown in Fig 4. Indeed, one should recall that meteorological covariates are the only part of the modelling chain that varies through time, and therefore the only source of information about rainfall seasonality.

We agree that the links between a single meteorological covariate and precipitation can change during the year for good physical reasons, but we argue that investigating the links between a set of several meteorological covariates and rainfall reduces the risk of misleading correlation. In practice, using this link between multiple meteorological covariates and rainfall features has been an effective way to encode rainfall seasonality in several past studies (e.g., [*Bárdossy and Plate*, 1992] [*Vrac et al.*, 2007a]

[*Benoit et al.*, 2020]). However, to avoid confusion, we will be more careful when describing the links between conditioning and seasonality in the revised manuscript.

*RC 1.18: Lines 189-192: if the models are to be used for downscaling climate model projections, there are other requirements as well e.g. that the covariates are well represented by climate models and capture the climate change signal (see, for example, Maraun and Widmann 2018, Section 11.5).*

AR 1.18: We fully agree with this comment, and we will add a paragraph in the discussion section to acknowledge it.

*RC 1.19: Lines 224 and 226: "GMM model" is a tautology. See also my previous comment about the allocation to the "most probable" state.*

AR 1.19: This is right, we will use GMM instead.

*RC 1.20: Lines 232-233: I don't understand what you're doing here. In lines 218-220 you said that you already estimated the parameters of $\psi$: why are you doing it again, therefore? Also, any simplistic method of bandwidth selection such as equation (7) is almost guaranteed to fail in a large proportion of applications: the authors' use of such an approach suggests that they don't really understand the potential pitfalls of such an approach – or, indeed, the availability of alternatives.*

AR 1.20: In lines 218-220 we estimate the parameters of $\psi$ for each day of the training dataset, which allows us to derive the empirical joint pdf of $p_0$, k and θ from observations. In contrast, in lines 232-233, we propose a non-parametric approach to sample this joint pdf and therefore simulate $p_0(d)$, k(d) and θ(d) for a given target day d. In a very schematic way we could therefore say that lines 218-220 correspond to inference and lines 232-233 correspond to simulation.
Regarding the choice of the simulation method, we also tested a parametric approach in which the parameters $p_0$, k and θ followed independent parametric distributions (we picked Gamma distributions for flexibility) that we inferred using a maximum likelihood approach. However, $p_0$, k and θ are correlated, which made this parametric approach leading to not fully satisfactory results. Hence the choice of a joint inference and simulation. The selection and inference of a flexible tri-variate distribution being cumbersome, we have decided to adopt a non-parametric approach. Although simple, this approach gives good results in practice as shown in the sections 3.2 and 3.3 where we can see that the marginal distributions of site-specific and island scale daily rain (i.e., the summary statistics that are the most impacted by $p_0$, k and θ) are both very well simulated by our model (Fig 4d, 5c, SM2.3d, SM2.4c).

*RC 1.21: Line 244: here, for the first time (I think) we discover that separate empirical copulas are being estimated for each day. What's the basis / justification for this? Are you not just resampling the original data, with a bit of smoothing? [actually this is mentioned on line 351, but I think it should be acknowledged upfront in the methodology].*

AR 1.21: It is true that this information (resampling of analogs) arrives a bit late in our paper, and it would have a better place in the model description (section 2.2) rather than in the model

implementation (section 2.3) section. In the revised paper, we will therefore introduce the resampling of analogs at the end of section 2.2, in the paragraph lines 147-152 (current numbering).

The justification for the adopted method – namely the resampling of separate empirical copulas for each day coupled with gamma transform and censoring – is that despite the classification of days into many rain types (e.g., 22 for O'ahu, which we justify hereafter), the intra-type variability of spatial patterns (and in turn of spatial copulas) remains significant. The best way to preserve this variability is therefore to resample daily analogs instead of using a single copula per type. As mentioned lines 350-353 and discussed lines 410-414, this approach indeed leads to results close to resampling the original data based on a preliminary classification into rain types, which we consider as a completely valid method for stochastic rainfall or weather generation in complex configurations (see e.g., [*Gangopadhyay and Clark*, 2005] [*Mezghani and Hingray*, 2009] [*Yiou*, 2014]), the innovation compared to previous studies being that our approach decouples spatial patterns and intensity in order to be able to create unobserved situations (which increases the variability rather than smooth the training dataset). We agree with reviewer#1 that this feature is an important part of our model, and we will mention it upfront in the methodology in the revised manuscript (i.e., by expanding the paragraph lines 147-152).

*RC 1.22: Line 281: do you really believe there are 22 distinct rainfall regimes on the island? You've only got 86 stations, so this classification doesn't seem to be reducing the spatial dimension as much as one might hope. As a slightly peripheral (but important) comment: Figure 3(b) will be inaccessible to the ~5% of male readers who suffer from red-green colourblindness.*

AR 1.22: Yes, we believe that 22 is a reasonable amount (even if not exact because of the many simplifications made during the classification process) for the number of distinct rainfall regimes on the island of O'ahu. This may seem a lot for modelers used to mid-latitude and/or continental climates, but we would like to take the opportunity of this answer to emphasize one more time the tremendous variability of rainfall and rainfall regimes in tropical islands, which in our opinion largely justifies the development of a dedicated stochastic rainfall model. We would like to mention here that the Hawaiian language has several hundreds of names to describe rain (e.g [*Akana and Gonzalez*, 2015]), which is at least one order of magnitude more than in European languages. In our opinion, this cultural difference indirectly illustrates the fact that far more distinct rainfall types exist in Hawai'i than for instance in Europe, and gives credibility to the high number of rain types identified in our study. A more quantitative justification for the proper dimension reduction of our classification is that the less complex (and smaller) dataset of Tahiti leads to an 11-type classification (supplementary material l41-44), which proves that our method can automatically adjust to the complexity of the dataset at hand (l381-384), and does not always produce very high numbers of rain types. We therefore believe that the 22 types obtained for O'ahu accurately describe the complexity and variability of rainfall observed on this island.

This being said, one should keep in mind that the selection of the number of components of the GMM, and in turn the number of rain types, is derived from an automatic model selection procedure based on a compromise between model parsimony and goodness-of-fit. Here we chose the Bayesian Information Criterion (BIC, [*Schwartz*, 1978]) to guide model selection, but other choices are possible (e.g., Aikaike Information Criterion - AIC), each criterion leading to a different ratio parsimony/goodness-of-fit. BIC has been chosen here because of its widespread use, and because it emphasizes the model parsimony (and therefore reduces the number of rain types) more than AIC. If one had good reasons (based on actual observations or previous knowledge about local rainfall

variability) to believe that a smaller number of rain types must be selected, one could adopt an even more "parsimonious" selection criterion, but we don't think this is required for the present dataset. Regarding the colors used in Fig 3b we agree with Reviewer#1, and we will therefore change the color scale for this figure and Fig SM2.1 to make them accessible to colorblind readers.

*RC 1.23: Lines 321-326: although it's good to test the model using a variety of measures, I'm not always convinced by the way that this has been done here – and I don't fully understand all aspects of the plot. What are the grey bands in column (b)? What do you mean by "each statistic is estimated as the median across the 50 realizations"? [I ask this because you have dashed lines indicating the quantiles, suggesting that you're showing the distribution rather than the median – in any case, it isn't appropriate to compare a single observation to the median of 50 simulations because they will have different statistical properties even if the model is correct]. There are published papers that deal with this issue correctly: the authors need to familiarise themselves with the literature. The issue is particularly acute in the Q-Q plots of column (d): if you really understood what a Q-Q plot represents, you'd realise that you don't need to duplicate the observations 50 times (I hope you haven't used the median of the simulations here as well ...). Similar comments apply to Figure 5.*

AR 1.23: There are several distinct concerns in this comment, and we will address them one at a time:
(16a) We share with Reviewer#1 the interest for multi-criteria model evaluation, and will try our best to make sections 3.2 and 3.3 easier to read and understand.
(16b) We made an error in the written caption of figure 3 (lines 321-326): in (a), (b), and (c), black refers to simulations and red refers to observations and not the opposite (the caption included in the figure was correct, but the description lines 321-324) was wrong. This probably caused confusion, and we would like to apologize for it.
(16c) We do not compare a single observation to the median of 50 simulations but rather compare distributions described by 3 quantiles: 10%, 50%, 90%. Hence the lower dashed red line is quantile 10% of observations, the continuous red line is quantile 50% of observations, and the upper dashed red line is quantile 90% of observations. If we would have simulated a single realization, the lower dotted black line (lower edge of the grey band) would have been the quantile 10% of this single realization, the continuous black line the quantile 50% of this realization, and the upper dotted black line its quantile 90%. However, we have simulated 50 realizations, and should therefore plot 50 lower dotted black lines, 50 central continuous black lines, and 50 upper dotted black lines, which would impede the readability of the figure. To improve on that point, we decided to summarize the information embedded into the 50 simulations of each quantile into a single black line, which we chose to be the median of each quantile evaluated through 50 realizations. This is what we meant by "*each statistic is estimated as the median across the 50 realizations*", the word "statistic" referring here to quantile 10%, quantile 50% and quantile 90%. To avoid confusion, we propose the following phrasing for the revised version of the paper: "*(b) Observed (red) and simulated (black) monthly rain accumulation. Dashed lines denote quantiles 10% and 90%, and solid lines denote the quantile 50%. In the case of simulations (black), for readability, we report in the figure only the median of each quantile (10%, 50%, 90%) instead of 50 simulated quantiles.*" To conclude on this point we would like to mention that such representation is not new in the literature related to stochastic rainfall models, and has been used for instance in Fig 4 of [*Peleg et al.*, 2017].
(16d) For q-q plots we didn't use the median of simulations but the 50 individual simulations as clearly indicated in the figure caption: "*Black dots line up vertically in q-q plots (d–e) because for each percentile, 50 simulations are compared to a single observation.*" (lines 325-326). We chose to duplicate observations in order to be able to draw one q-q plot for each realization, and therefore

show the variability between individual realizations. As for the previous point, we would like to mention here that such representation is not new in the literature related to stochastic rainfall models, and has been used for instance in Fig 1 of [*Allard and Bourotte*, 2015].

*RC 1.24: Lines 340-341: I don't know what are the likely applications of precipitation modelling in this location, but if the potential stakeholders include farmers then I think they may be justifiably sceptical of your claim that underestimation of persistence can be tolerated for mathematical convenience. Precipitation modelling is not a mathematical exercise, it relates to lives and livelihoods.*

AR 1.24: The likely applications the authors had in mind when designing this study were water resources management "from Mauka to Makai" (i.e., from mountain to the coast), hydro-power assessment at the island scale, and eco-hydrology of tropical island watersheds. That said, we agree that we should be more careful in the way we underline the drawbacks of our model and the associated limitations in term of applications (although we think that nothing was hidden here, we acknowledge a tactless wording). We therefore propose the following rephrasing: "Although the resulting errors are of low amplitude, they should be kept in mind and regarded as a limitation of the present model for applications requiring a precise estimation of rain persistence, as is the case for instance for crops models in semi-arid environments that can be found in some leeward areas of the target islands."

*RC 1.25: Lines 362-364: I'm not convinced by this dismissal of the curvature in the Q-Q plot of Figure 5(b). It is well-known that by compressing the tails of both observed and simulated distributions, Q-Q plots can make it hard to see discrepancies in the tails that may have substantial implications in applications. It would be helpful to consider alternative approaches to visualising this particular comparison (e.g. my guess is that if you were to plot the observed and simulated densities of proportions of dry gauges then you would be a bit more concerned about the simulation performance).*

AR 1.25: To avoid a one-sided representation of the statistics of interest, we display in the following figure both (1) the observed and simulated distributions of the statistics of interest (left column, as proposed by Reviewer#1) and (2) the associated q-q plots (right column). One can notice that the representation of the results in the form of PDFs confirms the conclusions derived from the q-q plots (section 3.3, l348-376), namely: (i) the overestimation of the frequency of low proportion of dry gauges (leading to the underestimation of the low quantiles), (ii) the overestimation of the maximum daily rain at the island scale (not distinguishable in the left column of this figure, but visible when zooming on the tail of the pdf, and of course on the q-q plot), and (iii) the good simulation of all other aspects of the evaluation statistics.

Regarding the imperfect simulation of the proportion of dry gauges, the effect is indeed easier to interpret when looking at the pdf, but the conclusions we draw from this plot are rather comforting. Indeed, one can see that the main discrepancy between simulations and observations is the overestimation of the frequency of proportions of dry rain gauges below 5%, which correspond in the present setting to less than 4 gauges over 86. In other words, our model tends to simulate rain at all 86 gauges when a very small number of gauges (<4) actually records no rain, leading to a slight 'drizzle effect' in space where rain is simulated at more locations than it should during generally wet days (more than 95% gauges record some rain). A careful examination of the spatial patterns of daily rain percentiles (Fig 5a) shows that the patterns of spatial intermittency of the percentiles are properly simulated, which suggests that the drizzle effect is randomly distributed amongst locations, which reduces its potential impact for applications. This is supported by the proper simulation of the marginal

distribution of rain accumulation (incl. zeros) at site-specific locations (Fig 4), which confirms that the drizzle effect at the island scale does not cause biases at the local scale.

All in all, we acknowledge an imperfect simulation of the proportion of dry gauges causing a drizzle effect in space at the island scale (which was already pointed out in lines 362-364 of our original manuscript) but we believe that this weakness is not dramatic for applications. However, to avoid any misuse of our model, this point will be discussed in more details in the revised version of the manuscript, and the figure below will be added to supplementary material to provide a more exhaustive visualization of our results.

[Figure]

*RC 1.26: Line 388: in what sense is the model structure "hierarchical"? It may be worth noting that "hierarchical modelling" has a precise technical meaning in statistics: this isn't what you are doing here. It's probably worth rephrasing for avoidance of ambiguity, therefore.*

AR 1.26: To avoid ambiguity and confusion, we will use "two-step model based on rain typing" instead of "hierarchical modelling" in the revised manuscript.

*RC 1.27: Line 392: here, the definition of weather types "based on rain features only" is offered as an apparent advantage of the proposed methodology. I would say that this is a distinct disadvantage because it ignores the physical processes that are operating. In my view, one of the key challenges in*

*stochastic weather generation is to find mathematically tractable ways of capturing the footprints of the fundamental physical processes: this remark from the authors suggests that they either haven't thought about this, or that they consciously disagree with me. At the very least, some more explanation is needed as to why this feature may be considered desirable and defensible.*

AR 1.27: We agree with Reviewer#1 that one of the key challenges in stochastic weather generation is to find statistical models able to capture the footprints of physical processes, but we don't see in which ways our rain-typing approach might be in contradiction with this aim. To go one step further, we would argue that if the main variable of interest is rainfall (which is the case of the present model), our approach allows a better identification of rainfall - climate interactions than a weather type approach, which would mix information about several meteorological variables during the classification step. In contrast, the choice of typing rainfall alone, and subsequently trying to relate rain type occurrence to meteorological covariates allows us to investigate (from a statistical point of view) the links that exist between the state of the atmosphere and the physical processes responsible for rain generation (incl. orographic effects). This is what we discuss in lines 395-401, and we think that this discussion is in line with the goal of statistically capturing physical processes stated by Reviewer#1.

*RC 1.28: Line 424: I disagree that the authors' model represents "a new tool". I think it represents a first step towards reinventing some existing tools (e.g. HMMs), but without acknowledging their existence (or perhaps not understanding what they're capable of).*

AR 1.28: We strongly believe that our model is "a new tool". We hope that our thorough responses to the above comments will have convinced readers and Reviewer#1 that (1) tropical island rainfalls are complex and very variable in both space and time, (2) these rains are different from the ones for which most stochastic rainfall models have been designed, (3) direct transposition of existing models is not easy, and (4) our modelling choices are reasonable and properly capture the main features of rainfall in tropical islands. To conclude, we think that independent from the novelty of the model itself, having a complete processing chain able to properly simulate daily rainfall in tropical islands (which we showed in our model assessment section 2.4, and in supplementary material 2.3) is already a valuable contribution and fully justifies our work.

---

## Author Comment (AC2)

Response to the comments of Prof. Geoff Pegram (Reviewer#2)

*Reviewer comment (RC 2.1): What a pleasure to read a hydrometeorological article bringing innovation and appropriate explanation in estimating, through resampling using valuable tools, spatially and temporarily highly variable precipitation on O'ahu, ready to be used in other counties with variable topography.   I have enjoyed the journey and am pleased to have reviewed it.*
*There is nothing much else to add, so I am returning my marked-up copy of the document. In addition, I am repeating a few of the less trivial remarks below my signature, which is my wont.*
*I recommend resubmission of a revised version in due course.*

Authors' response (AR 2.1): We would like to thank Prof. Pegram for his thoughtful review, and we are glad to read that our approach, initially designed for tropical islands, could find applications in different regions with complex topography.

*RC 2.2: L 71 A very useful introduction*

AR 2.2: Thank you, we're glad that you agreed with our choice to focus our introduction on the adaptation of stochastic rainfall models to the climate at hand.

*RC 2.3: L 89 storm (seasonal cyclones)*

AR 2.3: Ok, we will add this clarification in the revised manuscript.

*RC 2.4: L 101 Figure 1: Main features of rainfall observed over the island of O'ahu. Nice informative figure caption as are all of the others.*

AR 2.4: Thank you.

*RC 2.5: L 112 Figure 2: Overview of the structure of the stochastic rainfall model. Explain what you are doing more fully - also, how do you explain the almost perfect antithesis? What do the 'covariates' mean?  What do you mean by: "will be discussed in detail later" in line 110 above? - Where? Aha - you refer to Fig. 2 (not Figure 2) in the following paragraphs –confusing.*

AR 2.5: This is true that this figure occurs early in the paper (because it is an introductory figure), and therefore should have a more detailed caption. We expand and improve this caption in the revised manuscript. To answer the specific questions of Reviewer#2:
● This "almost perfect antithesis" is due to the nature of the covariates displayed here, specifically geopotential height at 950hPa (covariate 1) and temperature difference between 950hPa and 700hPa levels (covariate 2).
● Here "covariates" means atmospheric variables (e.g., pressure, temperature, humidity, wind) observed (or simulated) over the area of interest, and which are potential predictors for rainfall occurrence, intensity and spatial patterns.
● We will change this sentence line 110 to the following in order to avoid confusion: "Figure 2 summarizes the structure of the model, which will be discussed in detail in sections 2.2.2 to 2.2.4".

*RC 2.6: 2.2.2 Meta-Gaussian representation. 'Meta' can be used as an acronym for "most effective tactics available", so how you define it in this context, as it is unusual and I had to hunt for it in the web?*

AR 2.6: We agree, several terminologies co-exist in the literature (e.g., meta-Gaussian, trans-Gaussian, transformed-Gaussian), and meta-Gaussian may not be very frequent. We will define this term at the beginning of section 2.2.2 in the revised manuscript.

*RC 2.7: L 142  Eq. (2)  That's clever.*

AR 2.7: Thank you.

*RC 2.8: 286 Figure 3: Rain types identified for the island of Oʻahu. Figure 3(a) is too crushed for a person to read the busy text in comfort.  Please enlarge it and relocate 3(b) below. Your paper is short enough not to force the shrinkage.*

AR 2.8: Thank you, we will re-design Figure 3 to make it easier to read.

*RC 2.9: L 295  … (Fig. 3, rain types h–q)… Types h–q are labelled above by fading green captions an r-v are fading red - why?  A distraction - please change them to monochrome black.*

AR 2.9: Different colors have been used for labelling different rain types in order to allow for easy identification of rain type contributions in Figure 3b. But it is true that it can be distracting, and that it is redundant with the caption of Fig 3b. We will therefore follow this suggestion, and change these labels to monochrome black in the revised manuscript.

*RC 2.10: L 321 Figure 4: Ability of the model to simulate site-specific rain statistics on Oʻahu. I do not understand why you have a column (in (d)) of simulated values of near 125 mm/day and your simulated range is up to about 450 - same comment for the rest of the panels.  (a), (b) & (c) are easy to follow ...*

AR 2.10: This is because for the highest quantile (i.e., the 20-year maximum at the station of interest), the variability between the 50 realizations (i.e., independent simulations) becomes very large. For instance, in the first row of Fig. 4d, the observed 20-year maximum is 125mm/day and the simulated 20-year maximum varies between 125mm/day and 450mm/day depending on the realization, hence an overestimation of the 20-year maximum in this case. It should be noted that this large variability of simulated maximum is not seen as a negative result, but rather as an indication that this feature is poorly constrained in our model and should therefore be handled with care (see the discussion about the ability of our model to simulate extreme rainfalls, lines 370-376).

We will improve the caption of Figure 4d in the revised manuscript to make it easier to follow.

*RC 2.11: L 334 covariates properly capture rain type occurrence in a tropical marine climate. That's pretty convincing.*

AR 2.11: Thank you.

RC 2.12: L 355 Figure 5: Assessment of island-scale statistics simulation in O'ahu. Good work - the spread of mean and max daily rainfall are interesting - how do you account for the bias of the top-end in (d) with the max observed at 500mm? The text below does not comment on this.

AR 2.12: This bias means that our model overestimates the island-scale 20-year daily rain maximum, which is actually the main limitation of our model. This was discussed l 370-376 and restated in the conclusion lines 425-426. We will revise to make these points more clear.

RC 2.13: Comment on the Supplementary material: Figure SM2.4: Assessment of island-scale statistics simulation in Tahiti. This image is confirmation that the techniques were a success on Tahiti, transferred from your work on O'ahu. I suggest that you take a clip of the image [from (b) to (e)] to show us the value of your technique; the supplementary material is likely to be checked by few readers, so you need a teaser!

AR 2.13: Thank you for this suggestion. We will add a figure in section 3.4 to give an overview of the results obtained for the island of Tahiti.

RC 2.14: L 415 4.2 Concluding remarks. I think that this model will not be confined to tropical islands. I can see applications in my country, South Africa, where we have high coastal mountains in the East rising to 3000m flattening through dryer areas towards the North West. Check the figure from our work ; precip. max of 2000mm: "The gridded Mean Annual Precipitation averaged over each of the 1946 quaternary catchments in the region" in: Pegram GGS, Scott Sinclair and András Bárdossy (2016). New methods of infilling Southern African raingauge records enhanced by Annual, Monthly and Daily Precipitation estimates tagged with uncertainty. Water Research Commission, WRC Report No. 2241/1/15 ISBN.

AR 2.14: Thank you for sharing this insight and possible application. We are glad to read that our method can potentially be used in other areas with complex topography. It is true that orographic effects look impressive, in particular in Cape Town and Drakensberg areas. The main challenge we see to adapt the present framework to another climate (e.g., South Africa) is the selection of appropriate meteorological covariates to guide rain type occurrence.

RC 2.15: L 455 References. Please include doi's where available.

AR 2.15: Ok, we will add doi's every time this information is available.

RC 2.16: L 561 Scott, D. W. (2010), Scott's rule, WIREs Computational Statistics, 2, 497-502. doi:10.1002/wics.103. This find required a wild goose chase through the net but, eventually for your information, I found the original:- SCOTT, DAVID W. (1979). On optimal and data-based histograms. Biometrika, 66(3), 605–610. doi:10.1093/biomet/66.3.605

AR 2.16: Thanks a lot for this original reference. We will add it to the revised manuscript.

---

## Author Response (AR1)

Dear Editor and Reviewers,

Thank you for your detailed comments and suggestions about our manuscript entitled "Stochastic daily rainfall generation on tropical islands with complex topography".

Following your suggestions and the propositions of improvement we made in the public discussion, we thoroughly revised our manuscript. The main changes are the following:

1) A new paragraph has been added to the **Introduction** to explain why some features of tropical island rainfalls challenge existing stochastic rainfall generators, and therefore justify the need for a new modeling framework.
2) The section **2.2.1 Model overview** has been entirely re-written in order to better justify and contextualize our modeling choices.
3) A new section **3.2 Simulation of site-specific rainfall time series** has been created to display observed and simulated rainfall time series in addition to evaluation statistics, and therefore help readers better grasp the main features of Hawaiian rainfalls.
4) A new figure 7 has been added to the section **3.5 Model versatility** to better illustrate the flexibility of our model, and invites readers to check the case study of Tahiti presented in **Supplementary material 4**.

In addition to these major changes, every reviewer comment has been addressed as detailed in the following point-by-point responses. In these responses, "RC" (in blue) denotes a reviewer comment, "AR" (in black) denotes our response to the comment, and the text in italic corresponds to the modifications implemented in the revised manuscript. If no change has been implemented in the revised manuscript we repeat our answer from the public discussion preceded by the mention "Same response as in public discussion". In the point-by-point responses, the line numbers in the reviewer comments refer to those from the original manuscript, and the line numbers in the author responses refer to the revised manuscript.

We hope that our responses will address your concerns, and that our proposed improvements will meet your expectations.

Best regards,

Lionel Benoit, on behalf of the authors.
* * *
Response to the comments of Reviewer#1

Reviewer comment (RC 1.1): Some of the ideas in the paper are potentially interesting. However, the existing literature on weather generators (particularly for daily precipitation) is vast: to justify yet another new approach therefore, it is necessary to demonstrate that it improves on existing methods in some way (it could be the performance of the method, its ease of implementation, its range of applicability, its computational feasibility etc.). The authors do not provide any such demonstration, perhaps because they don't seem aware of the state of the art in the area.

Authors' response (AR 1.1): To demonstrate the need for a new modeling framework we added a paragraph in our introduction (l 66-80). This new paragraph details the specificities of rainfall in the context of tropical islands, and discusses how these features challenge existing modeling approaches based on literature review.

It reads as follow: *"Orographic effects lead to non-stationary (i.e., non-homogeneous) rain statistics in both space and time, which challenges most existing stochastic rainfall models [Nerini et al., 2017]. In the context of high tropical islands, the first difficulty arises from the long term patterns of orographic rain enhancement that create non-stationarities in space. Rainfall generators must therefore account not only for the correlation between locations [Leblois and Creutin, 2013] [Paschalis et al., 2013], but also for the location of the rain within the island. When orographic effects are directly related to topography, Generalized Linear Models (GLMs) have been leveraged to account for spatial rain patterns by linking model parameters to local topographic information such as altitude or slope aspect [Ambrosino et al., 2014] [Chandler, 2020]. In tropical islands however, the complexity of the rain-topography relationships (e.g., successive mountain ridges becoming drier at similar altitude) hinder the direct regression of model parameters on topographic data. An attractive option to overcome the non-uniqueness of topography-rainfall relationships is to interpolate the parameters of the rainfall model in space in order to let data inform non-stationarity [Kleiber et al., 2012] [Bennett et al., 2018]. This however increases model complexity, and therefore requires training datasets with high spatial resolution, which are not yet available in most target areas [Benoit et al., 2021]. Finally, the difficulty of accounting for orographic effects in the context of high tropical islands is further enhanced by the temporal variability of rainfall patterns (e.g., wet leeward slopes during Kona storms), which calls for the development of models able to identify and capture the main modes of variability of orographic effects over time."*

RC 1.2: Literature that seems particularly relevant includes that on Hidden Markov Models (e.g. Hughes et al. 1999; Ailliot et al. 2009): this uses the same basic idea of classifying each day on the basis of the joint spatial distribution of precipitation, but does so in what seems to be a more principled way than the present paper.

AR 1.2: The reason of choosing an explicit rain typing approach instead of a Hidden Markov Model is now discussed in section 2.2.1 (l145-156), which has been revised in depth to better justify and contextualize our modeling choices.

It reads as follows: *"The temporal variability of rain statistics and its relationships with the state of the atmosphere are modeled following a rain typing approach (Fig. 2a) [Ailliot et al., 2015] [Benoit et al., 2018b]. In this framework, days with similar rain statistics are first pooled together in a finite number of rain types, which represent summaries of island-scale daily rain statistics. Next, rain type occurrence is modeled by a Markov chain describing how rain types transition to each other [Bárdossy and Plate, 1991] [Wilby, 1994]. To preserve climatological consistency and model the influence of atmospheric circulation on orographic rain enhancement, rain type occurrence is conditioned to local meteorological covariates [Benoit et al., 2020] making the Markov chain non-homogeneous [Hughes and Guttorp, 1999] [Vrac et al., 2007]. It is important to note here that we do not use the widespread Hidden Markov Model (HMM) approach [Ailliot et al., 2009] [Greene et al., 2011] to introduce rain types in our stochastic rainfall model, but rather resort to a two-step approach in which rain types are first determined by a direct classification of rain gauge observations, and subsequently a statistical rainfall model is built for each rain type. This choice has been motivated by the use of non-parametric methods to model the distribution of rainfall conditional to rain types, which prevents the formulation of the likelihood of the full statistical model and in turn the use of a HMM to model rain types."*

RC 1.3: Moreover, approaches using generalised linear models with topographical indices as covariates (e.g. Ambrosino et al. 2014; Chandler 2020) address the issue of topographical variability directly: it's not obvious to me that such approaches would fail in a tropical island setting.

AR 1.3: The difficulty of applying Generalized Linear Models (GLMs) in the present context is now discussed in details in the revised introduction (l 66-80), cf AR 1.1.

RC 1.4: I am also surprised that the paper doesn't cite Maraun et al. (2010) which has become almost the canonical reference for anyone working in this area.

AR 1.4: In the revised manuscript we cite this reference in the first paragraph of the introduction, when we mention stochastic downscaling of climate projection as an application of stochastic rainfall models (l 41-44).

RC 1.5: In view of the concerns above, as well as some technical issues (see detailed comments below), I don't think the paper merits publication in its current form. To make the case, the authors need to

demonstrate that their approach improves on existing methods in some way as described above. Ideally, this would be done by carrying out an informed and fair comparison with a leading alternative method: if this isn't possible then the authors should explain why, and should offer some informed discussion of how their approach might reasonably be expected to compare.

AR 1.5: As mentioned above (AR 1.1 to AR 1.3), we improved the contextualization of our work and the justification of our modeling choices by (1) adding a new paragraph in the introduction to justify the need of a new stochastic rainfall model for tropical islands (l 66-80), and (2) completely rewriting the section 2.2.1 to explain the choices of the different components of our model (l 134-172).

In addition to better contextualization, model assessment has been improved to highlight the performance and the flexibility of the proposed model. Two main changes have been implemented:

1) Following Reviewer#1's advice to focus on realism of the simulations, we added the new section 3.2: Simulation of site-specific rainfall time series, which reads as follows:

*"Figure 4 displays stochastic rainfall generation outputs for the four rain gauges of Fig 1.a which experience different rainfall climatologies. Visual inspection of the simulated time series shows that our stochastic rainfall generator is able to simulate synthetic rains that are almost indistinguishable from the observed ones. The dry-wet ratio as well as the marginal distribution of daily rain accumulation seem properly replicated, except for the dry leeward gauge (first row in Fig. 4) where the 20-year maximum tends to be overestimated (see also Fig. 5.d). In terms of temporal variability, simulated time series properly capture the seasonal cycle visible for the two leeward gauges (rows 1-2 in Fig. 4), as well as the inter-annual variability visible for the two coastal gauges (rows 1 and 4 in Fig. 4). Finally, the rainfall generator captures the relatively steady behavior of the wet gauge located in the Koʻolau range (row 3 in Fig. 4).*

[Figure]

***Figure 4: Ability of the model to simulate realistic rainfall time series on Oʻahu.*** *(a) Target locations. (b) Observed rainfall time series for the period 1991-2011. (c-e) Simulated time series. For readability, only the three first realizations of the 50-member ensemble are displayed."*

2) Following Reviewer#2's advice to better highlight the flexibility of our model, we added the following figure to section 3,5 Model versatility:

[Figure]

*Figure 7: Overview of stochastic daily rainfall simulation for the island of Tahiti, French Polynesia. (a-e) Site-specific statistics: (a) Target locations. (b) Observed (red) and simulated (black) monthly rain accumulation. (c) Observed (red) and simulated (black) annual rain accumulation. (d) Q-q plot of daily rain percentiles. (e) Q-q plot of wet-spell duration percentiles. (f-i) Island-scale statistics: (f) proportion of dry gauges; (g–h) mean and max daily rain; (i) coefficient of variation.*

RC 1.6: Line 31: this dismissal of "detailed physical modelling of rain generation processes" seems quite one-sided and poorly informed. It is true that stochastic models are computationally faster than physical ones, but physical models do have their own advantages which are not acknowledged here.

AR 1.6: The previous formulation was indeed one-sided, and we therefore reformulated the comparison between physical and statistical models in a fairer way, with a focus on how physical models can (and should) inspire the design of stochastic models (l 45-50).

The new paragraph reads as follows: *"To enable fast and computationally efficient simulations, and thereby allow for investigation of rainfall variability and associated uncertainty through the simulation of large ensembles, stochastic rainfall generators adopt an empirical approach that bypasses the detailed physical modeling of rain generation processes [Bauer et al., 2015]. To avoid the pitfall of physically unrealistic simulations, stochastic rainfall models embed a significant part of our conceptual knowledge about rainfall behavior in their parameterization (i.e. they implement statistical relationships that reflect as closely as possible the physical processes at work)."*

RC 1.7: Line 79: what proportion of the data have been "gap-filled"? How is the gap-filling distributed across time and between stations? How was the filling done?

AR 1.7: More information about gap-filling, including the proportion of gap-filled data and a reference about the method used in practice have been added to the description of the study dataset (l 100-102).

The updated description reads as follows: *"The available rain gauge observation dataset consists of daily records from a network of 86 rain gauges spread over the island (Fig. 1a), and covers a 20-year period 1991–2011. It corresponds to a compilation of quality controlled and gap-filled daily observations [Longman et al., 2018]. Gap-filling was performed using the normal ratio method [Paulhus and Kohler, 1952], and only stations with less than 5% gap-filled data are kept for this study in order to minimize the impact of gap-filling on our results."*

RC 1.8: I will add that gap-filling in highly variable situations is, in my view, difficult and potentially dangerous because it will tend to underestimate variability. In my view therefore, for a stochastic rainfall model to be suitable for widespread use, it must be capable of handling incomplete datasets.

AR 1.8: [Same response as in public discussion]. We agree that incomplete datasets must be easily handled, that is why we selected (and implemented in our software) a pattern-preserving gap-filling method [*Oriani et al.*, 2020] for cases where no case-specific gap-filling has been performed (e.g., the Tahiti dataset, see supplementary material 2). This method has the advantage of being compatible with our modelling framework focusing on rainfall patterns. When a case-specific gap-filling has already been applied and validated (as is the case for the O'ahu dataset [*Longman et al.*, 2018]), we chose to base our study on gap-filled datasets.

RC 1.9: (in particular, what would you do if you needed to generate precipitation at a site for which you have no data? This requirement is common in many realistic applications). Lines 216-217 suggest that the proposed methodology cannot handle incomplete datasets: this is a serious limitation that needs to be acknowledged clearly and openly.

AR 1.9: In the revised manuscript, we acknowledge clearly and several times that the proposed model is multi-site (l 85, l 170, l 468, l 504). We also mention it as a limitation of the model, and acknowledge the need for spatial interpolation in case gridded outputs are required (l 507-509).
We therefore added the following paragraph to our concluding remarks: *"The main limitations are (i) the inaccurate simulation of extreme rainfalls, which calls for caution when using our model for flood risk assessment, and (ii) the restriction to multi-site simulation, which calls for an additional step of stochastic interpolation of the multi-site patterns when gridded outputs are required."*

RC 1.10: Lines 101-105: although the spatial variation in precipitation statistics initially seems dramatic here, a more considered inspection reveals that the statistics are more or less constant along NW-SE transects and that the predominant variation is basically along the axis of the trade winds. If one were to scale the seasonal cycles at the specimen locations to a common scale (e.g. proportion of annual rainfall falling in each month), I doubt whether they would be dramatically different. I don't see anything here that really challenges state-of-the-art multisite weather generators, therefore – and hence nothing that really necessitates the development of a new modelling framework.

AR 1.10: To avoid misinterpretation of Fig. 1, and in particular the wrong impression that rain statistics are almost constant along NW-SE transects, we added a sentence in the description of O'ahu's rainfalls to explain that successive mountain ranges along the trade winds become drier (l 108-110): *"Note some important details of this annual rainfall pattern, in particular that the rain maximum is observed leeward from the main crest of the Ko'olau range, and that the Ko'olau mountains are significantly wetter than the Wai'anae range despite higher elevation."*
In addition, we would like to mention here that Fig. 1a displays annual rain patterns to give a broad picture of orographic effects, but that daily patterns show way more variability than this temporally-integrated picture. This is why we discuss daily rainfall patterns when we describe fig. 1 in the main text (l 111-117). This aspect has been reinforced in the revised introduction (l 62-66) to clearly make the point that daily spatial patterns can dramatically diverge from the annual ones: *"To this first order quasi-static picture is added the important variability of daily rainfall patterns associated with processes ranging from synoptic-scale disturbances [Hopuare et al., 2018] [Longman et al., 2021] to large-scale atmospheric circulations [Hopuare et al., 2015] [Frazier et al., 2018] [Brown et al., 2020]. Large deviations from the long term rainfall patterns are thus observed, and usually dry leeward slopes can become the wettest part of the island, like during Kona storms (seasonal cyclones) in the Hawai'i archipelago [Caruso and Businger, 2006]."*
Regarding the difficulty of using GLMs in the context of tropical islands, this is now discussed in the revised introduction (l 70-76).

RC 1.11: Line 133: what's the justification for the formulation in equation (1)? It seems a bit ad hoc. Also, as defined, $Z_i$ isn't latent because it's directly connected to observable quantities.

AR 1.11: The choice of a Meta-Gaussian framework to model multi-site rainfall conditional to rain types as well as the associated terminology (incl. the latent field) is now discussed in details in section 2.2.1 Model overview (l 157-172).

The updated description of the Meta-Gaussian framework reads as follows: *"Conditional to each rain type, the distribution of rain across the island is modeled following a meta-Gaussian approach (also referred to as trans-Gaussian or transformed Gaussian) [Allard and Bourotte, 2015] [Baxevani and Lennartsson, 2015] [Papalexiou and Serinaldi, 2020]. In this framework, a latent field with standardized Gaussian marginal distribution is non-linearly transformed to match the marginal distribution of daily rainfall across the island, and the spatial dependencies of the latent field are used to encode the spatial distribution of rainfall (Fig. 2b-c). The latent field is often assumed to follow a multivariate-Gaussian distribution, which allows for a parsimonious modeling of the spatial dependencies using geostatistics [Lantuéjoul, 2002], and in turn the spatial interpolation of rain gauge observations [Benoit et al., 2018a]. In this study however, a careful investigation of inter-gauges spatial dependencies (Supplementary material 1) shows the inadequacy of the multivariate-Gaussian distribution to model the latent field. Instead, a non-parametric resampling of past analogs [Gangopadhyay and Clark, 2005] [Yiou, 2014] of the daily latent field is used to model spatial dependencies. Aside from the choice of the marginal distribution of the latent field (Gaussian vs. uniform), this model is comparable to the use of empirical spatial copulas [Bárdossy and Pegram, 2009] coupled with a parametric marginal distribution of non-zero rain accumulation. This approach has the advantage of faithfully reproducing the complex spatial distribution of rainfall due to orographic effects, but at the expense of the ability for spatial interpolation. The present model should therefore be regarded as a multi-site stochastic rainfall generator (and not a rain field generator), and the spatial interpolation of the simulated multi-site rainfall is kept out of the scope of this paper."*

RC 1.12: Line 142: similarly, what's the justification for the distance-weighting in equation (2)? This again seems a bit arbitrary, although it's certainly true that what is sometimes called the "spatial intermittence" problem is hard to resolve satisfactorily. Nonetheless, in my view it's not clear that the authors' proposal improves on that by, say, Stehlík and Bárdossy (2002) – another example of literature that they seem unaware of, or at least haven't considered as carefully as they should have done. And once again, it's not a "latent" field if it's defined in terms of interpretable quantities.

AR 1.12: Equation 2 is indeed an application-specific attempt to deal with spatial intermittence. This is now clearly-stated and better explained when we introduce Eq. 2 (l 188-191): *"The idea behind this application-specific solution to deal with the spatial intermittence of rainfall is that a location far from any wet gauge should remain dry even after combining the associated latent field with slightly different parameters of the transform function (Eq. 1-2) during the simulation step (Sect. 2.3.3)."*

RC 1.13: Lines 149-150: there's a claim here that the complex spatial distribution "prevents the use of a simple parametric form ... for the spatial copulas". It's not entirely clear to me what this means, but I assume it's something like: you can't find a "standard" spatial covariance model to represent the dependence structure between the $Z_i$. I'm prepared to believe this (although it could be partly an artefact of the artificial and deterministic distance-dependence structure of equation (2)), but it would be helpful to see some plots to justify it.

AR 1.13: Yes, we mean that the joint distribution of copulas between sites does not follow a stationary multi-variate Gaussian distribution, which implies that we cannot find easily a "standard" spatial covariance model to represent the dependence structure between the $Z_i$. This is now discussed l 161-170 (and reminded l 201-203): *"In this study however, a careful investigation of inter-gauges spatial dependencies (Supplementary material 1) shows the inadequacy of the multivariate-Gaussian distribution to model the latent field. Instead, a non-parametric resampling of past analogs [Gangopadhyay and Clark, 2005] [Yiou, 2014] of the daily latent field is used to model spatial dependencies."*

In addition, the following figure illustrating the non-multi-Gaussian behavior of the latent field has been added in Supplementary Material 1:

[Figure]

*Figure SM1.1: Bivariate scatterplots of latent values in the island of Oʻahu (same target stations as in Fig. 1 of the main paper). One can observe that bivariate spatial dependences are neither bivariate Gaussian (i.e., scatterplots do not have an elliptical shape) nor stationary (i.e., scatterplots are centered on different means, and have different spreads).*

RC 1.14: Lines 158-159: the ability to calculate principal components of the "latent" field shows that it isn't latent. See earlier comments on this.

AR 1.14: Please refer to our response AR 1.11 above.

RC 1.15: Sec 2.2.3: I find this "rain typing" approach, which seems to be derived from techniques that are popular in the machine learning community, to be rather clumsy compared with the much more principled approach taken by other authors using HMMs. Even if one were to accept that the approach is worth considering, the classification in lines 172-174 is inappropriate because it fails to account for the uncertainty in each day's weather type (e.g. if you had three states then you could find that the probabilities are always (0.34, 0.33, 0.33) in which case you would assign every day to state 1 which is clearly nonsense – exactly like the electoral system in some recently failed democracies).

AR 1.15: [Same response as in public discussion]. As mentioned in AR 1.2, we adopted 'explicit' rain types (i.e., not hidden) to enable the non-parametric steps of our framework, which are justified by the complexity of rainfall patterns in tropical islands. This choice of explicit rain types requires in turn finding an alternative to HMM for rain typing, and for this study we chose a Gaussian Mixture Model (GMM) classification. This classification approach is widely used for clustering (e.g., [*Fraley and Raftery*, 2002]), and has been applied successfully in the context of weather and rain typing (e.g., [*Vrac et al.*, 2007b] [*Benoit et al.*, 2018a]).

Regarding the classification (lines 172-174) we do not see anything wrong in our equations 4 and 5, which follow the usual GMM classification framework (e.g., [*Fraley and Raftery*, 2002]). We are not sure to understand the concern of reviewer#1. If we imagine that the probabilities of the three states (rain types) are 0.34, 0.33, 0.33 for all values of the feature-space, it means that the (multivariate

Gaussian, in our case 6-dimensional) distributions of these three types are almost identical, in which case the model will automatically detect that only one single type exists ($N_C = 1$). And therefore this problematic case of rain type non-identifiability will not occur.

RC 1.16: Line 179: there seems to be something wrong with this equation. If I understand correctly, it is something like a kernel estimate of the transition probabilities based on the Mahalanobis distance in the space of meteorological covariates; but if so, then summing the right-hand side over j should give 1 for any value of i or $MC_d$ (it's supposed to define a conditional probability distribution). It doesn't, at least if $\gamma_{ij}$ is a "baseline" transition probability (which isn't stated here, but seems to be the case based on a later description – if that's the case, then it's true that $\sum_j \gamma_{ij} = 1$ for all i but the exponential factor in line 179 messes things up).

AR 1.16: Thanks a lot for spotting this typo in equation 6, the sign '' must indeed be replaced by '$\propto$'. We corrected this error in the revised manuscript and added a reference to better contextualize non-homogeneous Markov Chains (l 228-230).

RC 1.17: Line 184: how does the conditioning inform seasonality? This is another slightly dangerous assertion: relationships between precipitation and meteorological covariates can themselves be seasonally varying, for good physical reasons (think precipitation and temperature in temperate latitudes - they are positively associated in winter but negatively associated in summer).

AR 1.17: To avoid misinterpretation, the revised manuscript mentions conditioning to meteorological covariates as an underline way to inform seasonality (l 241-242): *"the temporal variations of the covariates are correlated with variations in rain type occurrence (Supplementary Material 2), which indirectly informs the seasonality and interannual variability of rainfall patterns"*

RC 1.18: Lines 189-192: if the models are to be used for downscaling climate model projections, there are other requirements as well e.g. that the covariates are well represented by climate models and capture the climate change signal (see, for example, Maraun and Widmann 2018, Section 11.5).

AR 1.18: We fully agree with this comment, and in the revised manuscript the evaluation of the simulation of the meteorological covariates is mentioned as the main future work before being able to use the proposed model in the context of downscaling climate projections (l 514-517).
It reads as follows: *"An essential future investigation in this direction will be to assess how well GCMs simulate the set of meteorological covariates we selected to drive our stochastic rainfall generator, in particular the vertical temperature gradient used to inform shallow convection."*

RC 1.19: Lines 224 and 226: "GMM model" is a tautology. See also my previous comment about the allocation to the "most probable" state.

AR 1.19: "GMM model" has been changed to GMM throughout the paper.

RC 1.20: Lines 232-233: I don't understand what you're doing here. In lines 218-220 you said that you already estimated the parameters of ψ: why are you doing it again, therefore? Also, any simplistic method of bandwidth selection such as equation (7) is almost guaranteed to fail in a large proportion of applications: the authors' use of such an approach suggests that they don't really understand the potential pitfalls of such an approach – or, indeed, the availability of alternatives
.

AR 1.20: [Same response as in public discussion]. In lines 218-220 we estimate the parameters of $\psi$ for each day of the training dataset, which allows us to derive the empirical joint pdf of $p_0$, k and $\theta$ from observations. In contrast, in lines 232-233, we propose a non-parametric approach to sample this joint pdf and therefore simulate $p_0(d)$, $k(d)$ and $\theta(d)$ for a given target day d. In a very schematic way we could therefore say that lines 218-220 correspond to inference and lines 232-233 correspond to simulation.

Regarding the choice of the simulation method, we also tested a parametric approach in which the parameters $p_0$, k and $\theta$ followed independent parametric distributions (we picked Gamma distributions for flexibility) that we inferred using a maximum likelihood approach. However, $p_0$, k and $\theta$ are correlated, which made this parametric approach leading to not fully satisfactory results. Hence the choice of a joint inference and simulation. The selection and inference of a flexible tri-variate distribution being cumbersome, we have decided to adopt a non-parametric approach. Although simple, this approach gives good results in practice as shown in the sections 3.2 and 3.3 where we can see that the marginal distributions of site-specific and island scale daily rain (i.e., the summary statistics that are the most impacted by $p_0$, k and $\theta$) are both very well simulated by our model (Fig 4d, 5c, SM2.3d, SM2.4c).

RC 1.21: Line 244: here, for the first time (I think) we discover that separate empirical copulas are being estimated for each day. What's the basis / justification for this? Are you not just resampling the original data, with a bit of smoothing? [actually this is mentioned on line 351, but I think it should be acknowledged upfront in the methodology].

AR 1.21: The resampling of separate empirical latent fields as well as the link with empirical copulas is now described in details in section 2.2.1 (l 157-172). We therefore acknowledge the resampling of the original latent fields upfront in the methodology (l l165-166), and repeat this information when we describe the simulation step (l292-294). We also added some references to the description of analog-based simulation in order to better contextualize our simulation method (l 293).

RC 1.22: Line 281: do you really believe there are 22 distinct rainfall regimes on the island? You've only got 86 stations, so this classification doesn't seem to be reducing the spatial dimension as much as one might hope. As a slightly peripheral (but important) comment: Figure 3(b) will be inaccessible to the ~5% of male readers who suffer from red-green colourblindness.

AR 1.22: The large number of rain types is now discussed in details in the revised manuscript (l 332-337).

It reads as follows: *"Figure 3 displays the 22 rain types identified for O'ahu Island during the period 1991–2011. Although this number may seem high compared to the number of rain types inferred for mid-latitude continental climates (usually less than 10 types, see e.g. [Vrac et al., 2007] [Benoit et al., 2018b]), we believe that it reflects the tremendous variability of rains observed in the Hawai'i archipelago [Giambelluca et al., 2013], which led Hawaiians to use more than one hundred different words to describe rainfall [Akana and Gonzalez, 2015]."*

In addition, we changed the colors of figure 3(b) (as well as its equivalent for Tahiti in supplementary material, Fig SM4.1) to make it accessible to color-blind peoples.

RC 1.23: Lines 321-326: although it's good to test the model using a variety of measures, I'm not always convinced by the way that this has been done here – and I don't fully understand all aspects of the plot. What are the grey bands in column (b)? What do you mean by "each statistic is estimated as the median across the 50 realizations"? [I ask this because you have dashed lines indicating the quantiles, suggesting that you're showing the distribution rather than the median – in any case, it isn't appropriate to compare a single observation to the median of 50 simulations because they will have

different statistical properties even if the model is correct]. There are published papers that deal with this issue correctly: the authors need to familiarise themselves with the literature. The issue is particularly acute in the Q-Q plots of column (d): if you really understood what a Q-Q plot represents, you'd realise that you don't need to duplicate the observations 50 times (I hope you haven't used the median of the simulations here as well …). Similar comments apply to Figure 5.

AR 1.23: To avoid misunderstanding, we re-designed Fig. 5b-c and the associated caption (l 391-395). We also corrected the error in the caption where simulations and observations were inverted. The new figure is displayed hereafter:

[Figure]

***Figure 5: Ability of the model to simulate site-specific rain statistics on O'ahu.*** *(a) Target locations. (b) Observed (red) and simulated (black) monthly rain accumulation. Dashed lines denote quantiles 10% and 90%, and solid lines denote the quantile 50%. In the case of simulations (black), for readability, we report in the figure only the median of each quantile (10%, 50%, 90%) instead of 50 simulated quantiles. (c) Observed (red) and simulated (black) annual rain accumulation. In the case of simulations (black), for readability, we report in the figure only the median of the 50 simulations. (d) Q-q plot of daily rain percentiles. (e) Q-q plot of wet-spell duration percentiles. Black dots line up vertically in q-q plots (d–e) because for each percentile, 50 simulations are compared to a single observation.*

Regarding the relevance of our plots, we would like to mention that there are not new in the literature related to stochastic weather generators, and that Fig 5b is similar to e.g., Fig 4 of [*Peleg et al.*, 2017], and the q-q plots of Fig 5d-e and Fig 6b-e are similar to e.g., Fig 1 of [*Allard and Bourotte*, 2015].

RC 1.24: Lines 340-341: I don't know what are the likely applications of precipitation modelling in this location, but if the potential stakeholders include farmers then I think they may be justifiably sceptical of your claim that underestimation of persistence can be tolerated for mathematical convenience. Precipitation modelling is not a mathematical exercise, it relates to lives and livelihoods.

AR 1.24: The discussion of the underestimation of persistence has been modified to acknowledge its impact on applications (409-411): *"Although the resulting errors are of low magnitude, they should be kept in mind in case of applications requiring a precise estimation of rain persistence, for example crop models in semi-arid environments that can be found in some leeward areas of the target islands."*

RC 1.25: Lines 362-364: I'm not convinced by this dismissal of the curvature in the Q-Q plot of Figure 5(b). It is well-known that by compressing the tails of both observed and simulated distributions, Q-Q plots can make it hard to see discrepancies in the tails that may have substantial implications in applications. It would be helpful to consider alternative approaches to visualising this particular comparison (e.g. my guess is that if you were to plot the observed and simulated densities of proportions of dry gauges then you would be a bit more concerned about the simulation performance).

AR 1.25: To avoid one-sided interpretation of the cross-validation results, the results displayed as q-q plots in Fig 6 are also shown as pdfs in Supplementary material 3. Based on these two displays, we improved the discussion of the curvature in the q-q plot (l 433-438): *"Results show a slight underestimation of the low percentiles of the proportion of dry gauges (Fig. 6b), which corresponds to a slight overestimation of the frequency of proportions of dry rain gauges below 5% (i.e., 4 gauges over 86 in the present setting). That is, our model tends to simulate rain at all 86 gauges when a very small number of gauges (less than 4) actually records no rain, leading to a slight drizzle effect in space. A careful examination of the spatial patterns of daily rain percentiles (Fig 6a) shows that the patterns of spatial intermittency are properly simulated, which suggests that the drizzle effect is randomly distributed amongst locations, thus reducing its potential impact for applications."*

RC 1.26: Line 388: in what sense is the model structure "hierarchical"? It may be worth noting that "hierarchical modelling" has a precise technical meaning in statistics: this isn't what you are doing here. It's probably worth rephrasing for avoidance of ambiguity, therefore.

AR 1.26: To avoid ambiguity and confusion, we replaced "hierarchical modelling" by "two-step model based on rain typing" throughout the revised manuscript.

RC 1.27: Line 392: here, the definition of weather types "based on rain features only" is offered as an apparent advantage of the proposed methodology. I would say that this is a distinct disadvantage because it ignores the physical processes that are operating. In my view, one of the key challenges in stochastic weather generation is to find mathematically tractable ways of capturing the footprints of the fundamental physical processes: this remark from the authors suggests that they either haven't thought about this, or that they consciously disagree with me. At the very least, some more explanation is needed as to why this feature may be considered desirable and defensible.

AR 1.27: [Same response as in public discussion]. We agree with Reviewer#1 that one of the key challenges in stochastic weather generation is to find statistical models able to capture the footprints of physical processes, but we don't see in which ways our rain-typing approach might be in contradiction with this aim. To go one step further, we would argue that if the main variable of interest is rainfall (which is the case of the present model), our approach allows a better identification of rainfall - climate interactions than a weather type approach, which would mix information about several meteorological variables during the classification step. In contrast, the choice of typing rainfall alone, and subsequently trying to relate rain type occurrence to meteorological covariates allows us to investigate (from a statistical point of view) the links that exist between the state of the atmosphere and the physical processes responsible for rain generation (incl. orographic effects). This is what we discuss in lines 395-401, and we think that this discussion is in line with the goal of statistically capturing physical processes stated by Reviewer#1.

RC 1.28: Line 424: I disagree that the authors' model represents "a new tool". I think it represents a first step towards reinventing some existing tools (e.g. HMMs), but without acknowledging their existence (or perhaps not understanding what they're capable of).

AR 1.28: We hope that the thorough revision of our manuscript following the advices of the two referees have convinced Reviewer#1 that (1) tropical island rainfalls are complex and very variable in both space and time, (2) these rains are different from the ones for which most stochastic rainfall models have been designed, (3) direct transposition of existing models is not easy, and (4) our modelling choices are reasonable and properly capture the main features of rainfall in tropical islands. In practice, to better justify these modeling choices, a paragraph has been added to the introduction (l 66-80), and the model overview section that introduces the proposed model has been entirely rewritten (l 134-172).
* * *
Response to the comments of Prof. Geoff Pegram (Reviewer#2)

Reviewer comment (RC 2.1): What a pleasure to read a hydrometeorological article bringing innovation and appropriate explanation in estimating, through resampling using valuable tools, spatially and temporarily highly variable precipitation on O'ahu, ready to be used in other counties with variable topography.  I have enjoyed the journey and am pleased to have reviewed it.
There is nothing much else to add, so I am returning my marked-up copy of the document. In addition, I am repeating a few of the less trivial remarks below my signature, which is my wont.
I recommend resubmission of a revised version in due course.

Authors' response (AR 2.1): [Same response as in public discussion]. We would like to thank Prof. Pegram for his thoughtful review, and we are glad to read that our approach, initially designed for tropical islands, could find applications in different regions with complex topography.

RC 2.2: L 71 A very useful introduction

AR 2.2: [Same response as in public discussion]. Thank you, we're glad that you agreed with our choice to focus our introduction on the adaptation of stochastic rainfall models to the climate at hand.

RC 2.3: L 89 storm (seasonal cyclones)

AR 2.3: We added this clarification in the revised manuscript (l 64 and 114).

RC 2.4: L 101 Figure 1: Main features of rainfall observed over the island of O'ahu. Nice informative figure caption as are all of the others.

AR 2.4: [Same response as in public discussion].  Thank you.

RC 2.5: L 112 Figure 2: Overview of the structure of the stochastic rainfall model. Explain what you are doing more fully - also, how do you explain the almost perfect antithesis? What do the 'covariates' mean?  What do you mean by: "will be discussed in detail later" in line 110 above? - Where? Aha - you refer to Fig. 2 (not Figure 2) in the following paragraphs –confusing.

AR 2.5: The caption of figure 2 has been modified to explain the 'almost perfect antithesis': *"Figure 2: Overview of the structure of the stochastic rainfall model. (a) Meteorological covariates (here Geopotential at 950hPa and Temperature difference between 950hPa and 700hPa) driving the occurrence of rain types, which summarize daily rain statistics. (b) Latent field modeling of the spatial distribution of rainfall across the island. (c) Transform function linking latent values with actual rain accumulations. (d) Back-transform combining (b) and (c) to obtain daily rain simulations."*

In addition, we modified the confusing paragraph mentioned by Reviewer#2 in order to avoid ambiguity. The updated sentence reads as follows: *"Figure 2 summarizes the structure of the model, which is briefly introduced in the subsequent part of this subsection, and will be discussed in detail in sections 2.2.2 to 2.2.4."*

RC 2.6: 2.2.2 Meta-Gaussian representation. 'Meta' can be used as an acronym for "most effective tactics available", so how you define it in this context, as it is unusual and I had to hunt for it in the web?

AR 2.6: The description of the meta-Gaussian model has been improved, and the terminology is now better defined.

The updated description of the meta-Gaussian model reads as follows: *"Conditional to each rain type, the distribution of rain across the island is modeled following a meta-Gaussian approach (also referred to as trans-Gaussian or transformed Gaussian) [Allard and Bourotte, 2015] [Baxevani and Lennartsson, 2015] [Papalexiou and Serinaldi, 2020]. In this framework, a latent field with standardized Gaussian marginal distribution is non-linearly transformed to match the marginal distribution of daily rainfall across the island, and the spatial dependencies of the latent field are used to encode the spatial distribution of rainfall (Fig. 2b-c)."*

RC 2.7: L 142  Eq. (2)  That's clever.

AR 2.7: [Same response as in public discussion]. Thank you.

RC 2.8: 286 Figure 3: Rain types identified for the island of O'ahu. Figure 3(a) is too crushed for a person to read the busy text in comfort.  Please enlarge it and relocate 3(b) below. Your paper is short enough not to force the shrinkage.

AR 2.8: Figure 3 has been redesigned following these guidelines:

[Figure]

**Figure 3: Rain types identified for the island of Oʻahu.** *(a) Spatial distribution of daily rain and frequency of occurrence of each rain type. (b) Contribution of each rain type to the annual rain accumulation for a selection of 20 gauges spread throughout the island.*

RC 2.9: L 295  … (Fig. 3, rain types h–q)… Types h–q are labelled above by fading green captions an r-v are fading red - why?  A distraction - please change them to monochrome black.

AR 2.9: Figure 3 has been redesigned following these guidelines, cf updated figure above.

RC 2.10: L 321 Figure 4: Ability of the model to simulate site-specific rain statistics on Oʻahu. I do not understand why you have a column (in (d)) of simulated values of near 125 mm/day and your simulated range is up to about 450 - same comment for the rest of the panels.  (a), (b) & (c) are easy to follow …

AR 2.10: [Same response as in public discussion]. This is because for the highest quantile (i.e., the 20-year maximum at the station of interest), the variability between the 50 realizations (i.e., independent simulations) becomes very large. For instance, in the first row of Fig. 4d, the observed 20-year maximum is 125mm/day and the simulated 20-year maximum varies between 125mm/day and 450mm/day depending on the realization, hence an overestimation of the 20-year maximum in this case. It should be noted that this large variability of simulated maximum is not seen as a negative result, but rather as an indication that this feature is poorly constrained in our model and should therefore be handled with care (see the discussion about the ability of our model to simulate extreme rainfalls, lines 370-376).

RC 2.11: L 334 covariates properly capture rain type occurrence in a tropical marine climate. That's pretty convincing.

AR 2.11: [Same response as in public discussion]. Thank you.

RC 2.12: L 355 Figure 5: Assessment of island-scale statistics simulation in O'ahu. Good work - the spread of mean and max daily rainfall are interesting - how do you account for the bias of the top-end in (d) with the max observed at 500mm? The text below does not comment on this.

AR 2.12: [Same response as in public discussion], This bias means that our model overestimates the island-scale 20-year daily rain maximum, which is actually the main limitation of our model. This is discussed l 445-451 and restated in the conclusion lines 506-507.

RC 2.13: Comment on the Supplementary material: Figure SM2.4: Assessment of island-scale statistics simulation in Tahiti. This image is confirmation that the techniques were a success on Tahiti, transferred from your work on O'ahu. I suggest that you take a clip of the image [from (b) to (e)] to show us the value of your technique; the supplementary material is likely to be checked by few readers, so you need a teaser!

AR 2.13: Thank you for this suggestion. We added the figure 7 below as a teaser for the supplementary material 4.

[Figure]

Figure 7: Overview of stochastic daily rainfall simulation for the island of Tahiti, French Polynesia. (a-e) Site-specific statistics: (a) Target locations. (b) Observed (red) and simulated (black) monthly rain accumulation. (c) Observed (red) and simulated (black) annual rain accumulation. (d) Q-q plot of daily rain percentiles. (e) Q-q plot of wet-spell duration percentiles. (f-i) Island-scale statistics: (f) proportion of dry gauges; (g–h) mean and max daily rain; (i) coefficient of variation.

RC 2.14: L 415 4.2 Concluding remarks. I think that this model will not be confined to tropical islands. I can see applications in my country, South Africa, where we have high coastal mountains in the East rising to 3000m flattening through dryer areas towards the North West. Check the figure from our

work ; precip. max of 2000mm: "The gridded Mean Annual Precipitation averaged over each of the 1946 quaternary catchments in the region" in: Pegram GGS, Scott Sinclair and András Bárdossy (2016). New methods of infilling Southern African raingauge records enhanced by Annual, Monthly and Daily Precipitation estimates tagged with uncertainty. Water Research Commission, WRC Report No. 2241/1/15 ISBN.

AR 2.14: [Same response as in public discussion]. Thank you for sharing this insight and possible application. We are glad to read that our method can potentially be used in other areas with complex topography. It is true that orographic effects look impressive, in particular in Cape Town and Drakensberg areas. The main challenge we see to adapt the present framework to another climate (e.g., South Africa) is the selection of appropriate meteorological covariates to guide rain type occurrence.

RC 2.15: L 455 References. Please include doi's where available.

AR 2.15: Doi's have been added in the revised manuscript every time this information is available.

RC 2.16: L 561 Scott, D. W. (2010), Scott's rule, WIREs Computational Statistics, 2, 497-502. doi:10.1002/wics.103. This find required a wild goose chase through the net but, eventually for your information, I found the original:- SCOTT, DAVID W. (1979). On optimal and data-based histograms. Biometrika, 66(3), 605–610. doi:10.1093/biomet/66.3.605

AR 2.16: Thanks a lot for this original reference. We added it to the revised manuscript.

---

## Author Response (AR2)

Dear Editor Prof. Bravo de Guenni,

Thank you for your positive assessment of our revised manuscript entitled "Stochastic daily rainfall generation on tropical islands with complex topography".

To account for your last comments we modified our manuscript as follows:

*Editor comment: After your major revision of the manuscript hess-2021-453, I acknowledge that you have replied and addressed both reviewers' comments at a great extent. My main concern is still with your response to comment RC 1.23 about your Figure 5. You did not fully address all issues raised by the reviewer on this figure, like the meaning of the grey bands and the rather unusual QQ-plots of column d).*

Authors response: The caption of figure 5 has been improved (l 394-398) to explain that the grey bands in column (b) represent the quantile 10%-quantile 90% interval derived from simulations, and to better explain the Q-Q plots of columns (d) and (e).

*Editor comment: When you said that the "simulations properly reproduce site-specific marginal distributions of daily rain accumulation, except for the driest gauge where the 20-year maximum tends to be over-estimated", are you referring the first row of your figure?*

Authors response: Yes we are referring to Fig 5d, first row. To avoid confusion, we slightly modified the reference to this figure in the main text (l 414-415).

*Editor comment: For columns d) and column e), you are using the 50 simulated values, while for columns b) and c) you are using the median of the simulated values for comparison. Is this correct?*

Authors response: Yes, this is correct.

*Editor comment: Can you please also clarify the meaning of the black lines in Figures 6 and 7 represent?*

Authors response: Each black line connects the quantiles derived from a given simulation in order to help distinguish between simulations. We updated the caption of Figure 6 to explain it (l 431-434).

We hope that the above propositions of improvement will meet your expectations.

Best regards,
Lionel Benoit, on behalf of the authors.

---

## Author Response (AR3)

This is an accepted version of the manuscript. No changes have been done since the acceptance by the Editor Prof Bravo de Guenni on March 15$^{th}$ 2022.

Lionel Benoit on the behalf of the authors.